



# Satellite remote-sensing capability to assess tropospheric column ratios of formaldehyde and nitrogen dioxide: case study during the LISTOS 2018 field campaign

Matthew S. Johnson[1], Sajeev Philip[2], Rajesh Kumar[3], Aaron Naeger[4], Amir H. Souri[5], Jeffrey Geddes[6], Laura Judd[7], Scott Janz[8], John Sullivan[8]

[1]Earth Science Division, NASA Ames Research Center, Moffett Field, CA 94035, USA.
[2]Centre for Atmospheric Sciences, Indian Institute of Technology Delhi, Jia Sarai, Hauz Khas, New Delhi, Delhi 110016, India.
[3]Research Applications Laboratory, National Center for Atmospheric Research, Boulder, CO 80305, USA.
[4]Short-term Prediction Research and Transition Center, University of Alabama in Huntsville, Huntsville, AL 35805, USA.
[5]Atomic and Molecular Physics (AMP) Division, Center for Astrophysics | Harvard & Smithsonian, Cambridge, MA, USA.
[6]Earth and Environment Department, Boston University, Boston, MA, 02215, USA.
[7]NASA Langley Research Center, Hampton, VA 23681, USA.
[8]NASA Goddard Space Flight Center, Greenbelt, MD 20771, USA.

*Correspondence to*: Matthew S. Johnson (matthew.s.johnson@nasa.gov)





**Abstract.** Satellite retrievals of tropospheric column formaldehyde (HCHO) and nitrogen dioxide ($NO_2$) are

frequently used to investigate the sensitivity of ozone ($O_3$) production to concentrations and emissions of nitrogen oxides ($NO_x$) and volatile organic carbon compounds (VOCs). Space-based remote-sensing information of chemical proxies for $NO_x$ (i.e., $NO_2$) and VOCs (i.e., HCHO), in particular the ratios of tropospheric column HCHO and $NO_2$ (FNRs), provide insight into the non-linear relationship of $O_3$ formation in the lower troposphere. Ultraviolet–visible (UV/VIS) satellite spectrometers such as the Ozone Monitoring Instrument (OMI) and TROPOspheric Monitoring

Instrument (TROPOMI) are capable of providing FNR information with high spatiotemporal coverage, yet a recent study suggested that the biases and noise of satellite retrievals are the largest source of uncertainty for applying satellite-derived FNRs to better understand $O_3$ production sensitivities. To quantify, and inter-compare, the uncertainties in two of the most commonly-applied satellite sensors to investigate $O_3$ production sensitivities, we evaluated OMI and TROPOMI retrievals of $NO_2$ and HCHO tropospheric columns, and resulting FNRs, using

Geostationary Trace gas and Aerosol Sensor Optimization (GeoTASO) and GEO-CAPE Airborne Simulator (GCAS) airborne remote-sensing data taken during the Long Island Sound Tropospheric Ozone Study 2018 (LISTOS 2018).

Compared to suborbital remote-sensing observations of tropospheric column $NO_2$ and HCHO, the accuracy of OMI (using both the National Aeronautics and Space Administration (NASA) version 4 and the Quality Assurance for Essential Climate Variables (QA4ECV) retrieval algorithms) and TROPOMI were magnitude-dependent with high

biases (i.e., satellite tropospheric columns > suborbital tropospheric columns) in clean/background environments and a tendency towards a low bias (i.e., satellite tropospheric columns < suborbital tropospheric columns) in moderate to polluted regions. Campaign-averaged $NO_2$ median biases for OMI, using both the NASA and QA4ECV algorithms, were similar at $0.4\pm4.1 \times 10^{15}$ molecules $cm^{-2}$ (6.3%) and $0.4\pm4.5 \times 10^{15}$ molecules $cm^{-2}$ (6.8%), respectively. TROPOMI retrievals of $NO_2$ had a campaign-averaged median bias of $-0.3\pm3.7 \times 10^{15}$ molecules $cm^{-2}$ (-4.8%) and

$0.3\pm3.3 \times 10^{15}$ molecules $cm^{-2}$ (5.8%) when averaged at finer ($0.05° \times 0.05°$) and coarser ($0.15° \times 0.15°$) spatial resolution. The three satellite products (NASA OMI, QA4ECV OMI, and TROPOMI) differed more when evaluating tropospheric column HCHO retrievals. Noise in the HCHO retrievals, likely due to low signal-to-noise ratios and the fact the UV/VIS measurement sensitivity at shorter wavelengths used in HCHO retrievals are low in the troposphere, resulted in low correlations and high oscillation/variability in bias (bias standard deviation) in all three satellite

products, with campaign-averaged median biases of $5.1\pm7.8 \times 10^{15}$ molecules $cm^{-2}$ (38.7%), $2.3\pm8.9 \times 10^{15}$ molecules $cm^{-2}$ (17.3%), $1.9\pm6.7 \times 10^{15}$ molecules $cm^{-2}$ (12.9%), and $2.9\pm4.9 \times 10^{15}$ molecules $cm^{-2}$ (23.1%) for NASA OMI, QA4ECV OMI, and TROPOMI at finer and coarser spatial resolution, respectively. Spatially-averaging TROPOMI tropospheric column HCHO, along with $NO_2$ and FNRs, to coarser resolutions similar to OMI native pixel size proved to reduce the bias standard deviation of the retrieval data. While large median biases, and enhanced variability in bias,

were derived for HCHO, errors in both $NO_2$ and HCHO tropospheric columns tended to offset as all three satellite products compared well to observed FNRs with campaign-averaged median biases from NASA OMI, QA4ECV OMI, and TROPOMI of 0.4±3.8 (11.0%), -0.2±3.3 (-5.4%), and 0.4±2.3 (13.0%), respectively. While satellite-derived FNRs had minimal campaign-averaged median biases, the statistical analysis shows that all satellite FNR values still had large bias standard deviation due to unresolved errors in satellite retrievals of HCHO. This result is important as

accurate retrievals (minimal median biases) of FNRs from satellites do not suggest the accuracy of the underlying



proxy species. The reduction in noise in satellite retrievals of HCHO with additional calibration and improved sensor design and/or improved a priori information of the vertical profiles of HCHO in the troposphere to avoid the impact of the low measurement sensitivity in the shorter UV/VIS wavelengths used to retrieve HCHO is critical for reducing unresolved biases in satellite retrievals of FNRs. Furthermore, this work demonstrates the large impact of a) a priori

vertical profiles of $NO_2$ and HCHO for calculations of Air Mass Factors in tropospheric column trace gas retrievals in both OMI and TROPOMI, b) spatiotemporal averaging to increase signal-to-noise, and c) different retrieval algorithms on retrieval errors. Finally, the novel diurnal information of tropospheric FNRs that is expected to be provided by the upcoming NASA geostationary sensor Tropospheric Emissions: Monitoring of Pollution (TEMPO) is investigated and compared to low earth orbiting sensors currently applied to investigate tropospheric FNRs.






## 1 Introduction

Tropospheric ozone ($O_3$) is a harmful pollutant and near-surface concentrations of this species have detrimental impacts on human- and environmental-health (Kampa and Castanas, 2008; Van Dingenen et al., 2009). The production and destruction rates of tropospheric $O_3$ are controlled by complex chemical reactions involving the primary precursor species of nitrogen oxides ($NO_x$ = nitric oxide and nitrogen dioxide ($NO + NO_2$)) and volatile organic compounds (VOCs) (Sillman, 1999; Lelieveld and Dentener, 2000). It is critical to understand precursor species emissions and subsequent atmospheric chemistry controlling surface-level $O_3$ production rates since the United States (US) Environmental Protection Agency (EPA) designs and enforces concentration limits of criteria pollutants (e.g., $O_3$, $NO_2$, carbon monoxide, particulate matter, and sulfur dioxide) under the National Ambient Air Quality Standards (NAAQS). The current NAAQS for $O_3$ requires that 3-year averaged annual fourth-highest daily maximum 8-hour mean concentrations be $\leq 70$ ppb (US EPA, 2015). To reduce and maintain surface-level $O_3$ concentrations below NAAQS thresholds, many regions have designed and implemented emission control strategies of precursor species. To design effective emission reduction strategies, knowledge about the non-linear sensitivity of $O_3$ formation to $NO_x$ and VOCs is critical (Crutzen, 1973; Sillman, 1999). Based on the relative concentrations of $NO_x$ and VOCs, the formation of $O_3$ is sensitive to perturbations of either $NO_x$ ($NO_x$-limited regimes) or VOC emissions ($NO_x$-saturated or VOC/radical-limited regimes). These $O_3$ sensitivity regimes are separated by a transitional regime where $O_3$ formation is sensitive to changes in both $NO_x$ and VOC emissions.

To understand the non-linear relationship of $O_3$ formation to $NO_x$ and VOC emissions in complex chemical environments (e.g., polluted regions and areas of heterogenous concentrations/emissions of $NO_x$ and VOCs), spatiotemporally dense in situ measurements or airborne remote-sensing observations of precursor species concentrations and chemical reactivity are desired (e.g., Souri et al., 2020). Since these measurements are often spatiotemporally sparse, to supplement the time and space void of these observations, thoroughly evaluated model simulations can be applied. However, the accuracy of chemical transport models (CTMs) is highly dependent on inputs such as emission inventories, simulated meteorology, chemistry mechanisms, and removal processes all of which have varying levels of uncertainty. These model uncertainties can directly impact the understanding of the non-linear relationship of $O_3$ formation when using these simulated data (e.g., Choi and Souri, 2015). In the absence of accurate in situ measurements or high spatiotemporal suborbital remote-sensing information of chemical proxies for $NO_x$ (i.e., $NO_2$) and VOCs (i.e., formaldehyde (HCHO)), satellite retrievals of these species have also been demonstrated to provide insight into the $O_3$-$NO_x$-VOC relationship (Tonnesen and Dennis, 2000; Martin et al., 2004; Duncan et al., 2010; Souri et al., 2017; Jin et al., 2017, 2020). The ratio of HCHO to $NO_2$ concentrations (hereinafter FNR) has been demonstrated to provide information to monitor the local sensitivity of $O_3$ production from the chemical loss of $HO_2+RO_2$ ($LRO_x$) and chemical loss of $NO_x$ ($LNO_x$) controlling $O_3$-$NO_x$-VOC chemistry (Tonnesen and Dennis, 2000; Kleinman et al., 2001).

Multiple past and current space-based spectrometers have the capability to retrieve simultaneous $NO_2$ and HCHO tropospheric columns including Global Ozone Monitoring Experiment (GOME, Martin et al., 2004), GOME-2 (Choi et al., 2012), Ozone Monitoring Instrument (OMI, Duncan et al., 2010), and TROPOspheric Monitoring Instrument (TROPOMI, Chan et al., 2020, Souri et al., 2021). In addition to these low earth orbiting satellites,





Tropospheric Emissions: Monitoring of Pollution (TEMPO) is an upcoming National Aeronautics and Space Administration (NASA) geostationary satellite mission which will retrieve hourly $NO_2$ and HCHO tropospheric columns over North America (Zoogman et al., 2017; Chance et al., 2019). This geostationary sensor over North America is part of a constellation of air quality spaceborne sensors including the Geostationary Environment Monitoring Spectrometer (GEMS) instrument onboard the Korean Aerospace Research Institute GEO-KOMPSAT-2B satellite (Kim et al., 2020) and the European Space Agency (ESA) Sentinel-4 mission (ESA, 2017). Satellite retrievals of $NO_2$ and HCHO have been applied to determine the sensitivity of $O_3$ formation to $NO_x$ and VOC emissions at coarse spatial and temporal scales (e.g., Martin et al., 2004; Duncan et al., 2010) to finer spatiotemporal scales and focusing on long-term trends (e.g., Choi et al., 2012; Jin and Holloway, 2015; Choi and Souri, 2015; Schroeder et al., 2017; Souri et al., 2017; Jin et al., 2017, 2020). However, uncertainties remain in how accurately satellites can retrieve information needed to study surface-level or planetary boundary layer (PBL) $O_3$-$NO_x$-VOC relationships. These uncertainties stem from a) the exact thresholds of FNRs that separate $NO_x$-limited, transition, and VOC-limited regimes, b) the ability of tropospheric column retrievals to represent PBL chemical composition for air quality purposes due to variability in the vertical structure of $NO_2$ and HCHO concentrations and satellite sensitivity throughout the entire troposphere, c) whether HCHO is an effective proxy for total VOC reactivity, d) satellite spatial representation errors, and e) the accuracy/uncertainty of satellite retrievals of tropospheric column HCHO and $NO_2$. Of all these sources of uncertainty, mean/median and random biases due to noise in satellite retrievals of tropospheric column HCHO and $NO_2$ may be the largest source of error for retrieving FNRs using satellite sensors (Souri et al., 2022a). Therefore, it is vital to accurately define the level of errors/biases associated with satellite sensors to understand the capability of using this spatiotemporally-dense data source for investigating the impact of $NO_x$ and VOC emission perturbations on $O_3$ chemistry.

This study is designed to demonstrate the effectiveness of two frequently applied satellites for evaluating $O_3$-$NO_x$-VOC relationships (i.e., OMI and TROPOMI) to accurately retrieve tropospheric HCHO and $NO_2$ column concentrations and the subsequent tropospheric column FNRs. OMI and TROPOMI retrievals have been evaluated in numerous studies (e.g., Judd et al., 2020; Vigouroux et al., 2020; Zhu et al., 2020; Lamsal et al., 2021), typically focusing on a specific sensor and species (e.g., evaluating OMI or TROPOMI and $NO_2$ or HCHO separately); however, not for the accuracy to retrieve tropospheric column FNRs. Here we validate OMI and TROPOMI retrievals of HCHO and $NO_2$, and subsequent FNRs, with airborne spectrometer data obtained during the Long Island Sound Tropospheric Ozone Study 2018 (LISTOS 2018) field campaign conducted during the summer of 2018 in the northeast region of the US. Furthermore, this work demonstrates the additional information of tropospheric FNRs that is expected to be provided by the upcoming NASA geostationary sensor TEMPO. The manuscript is designed as follows. Section 2 presents the satellite, airborne remote-sensing, model data. and evaluation techniques applied in this study. The results are reported in Sect. 3 and the final conclusions are presented in Sect. 4.

## 2 Methods

This study focuses on the spatial domain and time period (June 25 to September 6, 2018) of the LISTOS 2018 (https://www.nescaum.org/documents/listos; https://www-air.larc.nasa.gov/missions/listos/index.html) field



campaign. This campaign was chosen due to the overlap of the TROPOMI and OMI missions, the availability of
airborne spectrometer retrievals (i.e., Geostationary Trace gas and Aerosol Sensor Optimization (GeoTASO) and
GEO-CAPE Airborne Simulator (GCAS)) of tropospheric column HCHO and $NO_2$ which are effective satellite
validation data (e.g., Judd et al., 2020), and the large spatiotemporal coverage of the airborne spectrometer data. Many
studies have applied stationary sources of ground-based remote-sensing data to validate OMI and TROPOMI (e.g.,
MAX-DOAS, FTIR, Pandora); however, using the airborne GeoTASO and GCAS products allows for the evaluation
of the satellite retrievals in variable environments (i.e., clean/background to heterogenous/polluted regions) in the
same day. The rest of this section describes the remote-sensing and model data applied in this study for evaluation of
tropospheric column HCHO and $NO_2$ from OMI and TROPOMI.

## 2.1 OMI remote-sensing products

The Dutch-Finnish nadir viewing spectrometer OMI, onboard the polar-orbiting NASA Aura satellite, which was
launched in 2004, is an ultraviolet–visible (UV/Vis) spectrometer (Levelt et al., 2006). Retrievals are made from three
wavelength channels between 260 to 510 nm (UV-1: 264 to 311 nm, UV-2: 307 to 383 nm, Vis: 349 to 504 nm). Aura-
OMI has a local equatorial overpass time of ~13:45 with nearly-complete daily global surface coverage due to the
large ~2,600 km swath width. Level-2 (L2) tropospheric vertical column density (VCD) OMI $NO_2$ retrievals from the
NASA version 4 standard product (OMNO2; Lamsal et al., 2021) and the NASA operational OMI HCHO version 3
product using the Smithsonian Astrophysical Observatory (SAO) retrieval algorithm (OMHCHO; González Abad et
al., 2015, 2016) were applied in this study. To investigate the impact of different retrieval algorithms, we also apply
tropospheric column OMI $NO_2$ and HCHO data derived in the Quality Assurance for Essential Climate Variables
(QA4ECV) project (see Sect. 2.1.2).

Starting in 2007, OMI experienced a field-of-view blockage known as the "row anomaly" which affects the
data quality at all retrieval wavelengths for some rows (Dobber et al., 2008; Schenkeveld et al., 2017). The row
anomaly in $NO_2$ retrieval is avoided in this study by filtering out rows/pixels flagged by the row anomaly detection
algorithm. The postprocessing bias correction approach using the reference sector method for OMI HCHO is applied
here and corrects for the row anomaly in HCHO data (De Smedt et al., 2015). OMI data also has systematic biased
retrievals in a striped pattern running in 60 cross-track field-of-views. A "de-striping" correction is already applied to
the $NO_2$ data (Boersma et al., 2011) and the reference sector method corrects for these artifacts in the HCHO data (De
Smedt et al., 2015; González Abad et al., 2015; Zara et al., 2018).

### 2.1.1 OMI – NASA OMNO2 and OMHCHO

The primary OMI data applied in this study are the L2 tropospheric VCD OMNO2 and OMHCHO retrievals provided
at ~13 km × 24 km near nadir to ~24 km × 160 km towards the edge of the swath. Lamsal et al. (2021) describes the
OMNO2 retrieval algorithm in detail and is explained here only briefly (referred to as NASA OMI $NO_2$ throughout).
The NASA OMI $NO_2$ retrieval uses a differential optical absorption spectroscopy (DOAS) approach, with a fitting
window between 405 and 465 nm, to derive slant column densities (SCD) of $NO_2$. Tropospheric $NO_2$ columns are
separated from the entire atmospheric column using an observation-based stratosphere–troposphere separation scheme



described in Bucsela et al. (2013). Tropospheric SCDs are then converted to tropospheric VCDs using an Air Mass
Factor (AMF) calculated with a radiative transfer model and simulated atmospheres from a CTM. The AMF is an
integrated product of scattering weights (SWs) and trace gas profile shapes (Palmer et al., 2001). Tropospheric AMFs
are calculated in NASA OMI $NO_2$ retrievals using monthly-averaged a priori profiles from the NASA Global
Modelling Initiative (GMI) model at $1° \times 1.25°$ spatial resolution, clouds from the OMI $O_2$–$O_2$ algorithm (Vasilkov
et al., 2018), and surface albedo from geometry-dependent surface Lambertian equivalent reflectivity (GLER) data

(Vasilkov et al., 2017; Qin et al., 2019; Fasnacht et al., 2019). The uncertainty of the tropospheric NASA OMI $NO_2$
product has been shown to vary with cloudiness and pollution concentrations and is in the range of ~20% to ~60%
(Bucsela et al., 2013), with contributions from errors in spectral fitting, stratospheric correction, and AMF calculations.

González Abad et al. (2015, 2016) describes the OMHCHO retrieval algorithm in detail (referred to as NASA
OMI HCHO throughout). Briefly, retrievals of HCHO SCDs are obtained by spectrally fitting OMI radiances using

the basic optical absorption spectroscopy (BOAS) method (Chance, 1998) with a fitting window between 328.5 and
346.0 nm. Then, like NASA OMI $NO_2$ retrievals, SCDs are converted to VCDs applying derived AMFs using GEOS-
Chem a priori profiles, cloud information (Martin et al., 2002; Acarreta et al., 2004), and surface albedo data (Vasilkov
et al., 2014). Finally, postprocessing across-track bias corrections are applied by comparing daily HCHO VCDs with
background VCDs simulated with the GEOS-Chem CTM over a clean region (known as the reference sector). The

uncertainty of the HCHO product has been shown to vary with pollution concentration ranging from ~45% to ~105%
with largest contributions from the spectral fitting and AMF calculations (González Abad et al., 2015, 2016).

### 2.1.2 OMI – QA4ECV $NO_2$ and HCHO

For comparison to the NASA OMI retrieval products, we inter-compared and evaluated OMI $NO_2$ and HCHO
retrievals from the QA4ECV project (www.qa4ecv.eu). Retrievals from the QA4ECV $NO_2$ version 1.1 and QA4ECV

HCHO version 1.2 data products are applied in this study and are provided daily at the same spatial resolution as the
NASA OMI products (~13 km × 24 km near nadir to ~24 km × 160 km towards the edge of the swath). Zara et al.
(2018) describes the QA4ECV OMI $NO_2$ and HCHO slant column retrievals and Boersma et al. (2018) and De Smedt
et al. (2018) describe the entire QA4ECV OMI $NO_2$ and HCHO retrieval algorithms, respectively, in detail. They are
summarized here briefly.

QA4ECV retrievals of $NO_2$ SCDs are obtained by linear fits of optical depths to the observed optical depth
using the DOAS technique with a fitting window between 405 and 465 nm (Boersma et al., 2018). While the QA4ECV
$NO_2$ retrieval is based on DOAS methods, it differs from the NASA OMI $NO_2$ retrieval in many of the retrieval steps
(Compernolle et al., 2020). For instance, the OMNO2 retrieval algorithm uses non-linear fits of modelled reflectance
to the observed reflectance. Furthermore, NASA OMI $NO_2$ uses an iterative fitting procedure compared to a

simultaneous fitting applied in QA4ECV. To calculate tropospheric AMFs, the QA4ECV $NO_2$ retrieval algorithm uses
the same surface albedo (Kleipool et al., 2008) and cloud products (Veefkind et al., 2016) as the previous NASA OMI
$NO_2$ version 3 data (see Lamsal et al., 2021); however, uses daily a priori profiles from the TM5 CTM at $1° \times 1°$
spatial resolution. Tropospheric VCDs of $NO_2$ are separated from the entire column using output from the global TM5
assimilation model in the QA4ECV $NO_2$ retrieval. For detailed information on the differences in spectral fitting





between the NASA OMI $NO_2$ and QA4ECV $NO_2$ retrieval algorithms we refer you to Zara et al. (2018). For details about differences between AMF calculations in the NASA and QA4ECV OMI algorithms see Lorente et al. (2017). QA4ECV $NO_2$ data have been shown to perform relatively well in clean to moderately polluted regions and have a low bias in highly polluted regions (Compernolle et al., 2020). Retrievals of QA4ECV HCHO SCDs are conducted in a similar manner to QA4ECV $NO_2$ using the DOAS technique and optical depths with a fitting window between 328.5

and 346.0 nm (Zara et al., 2018; De Smedt et al., 2018). QA4ECV HCHO retrievals show minimal bias in clean to moderately polluted regions and low biases in polluted locations (e.g., De Smedt et al., 2021).

## 2.2 TROPOMI remote-sensing products

The TROPOMI hyperspectral spectrometer (including eight bands in the UV, VIS, near-infrared, and shortwave infrared wavelengths) is onboard the Sentinel-5 Precursor (S5P) satellite developed by the ESA which was launched

in October 2017. TROPOMI is in orbit with a similar local equatorial overpass time (local time ~13:30) as OMI. TROPOMI has a swath width of ~2,600 km and a ground pixel size of 3.5 km × 7.0 km at nadir during the time of this study (since August 6, 2019 TROPOMI data is available at 3.5 km × 5.5 km) which is >12 times finer than OMI. TROPOMI retrievals have been used in numerous recent studies investigating processes controlling $NO_2$ concentrations and trends (e.g., Goldberg et al., 2021) and FNRs (Wu et al., 2022), taking advantage of the high

spatiotemporal resolution of the sensor, along with being validated thoroughly (e.g., Judd et al., 2020; De Smedt et al., 2021). The high spatial resolution information provided by TROPOMI, compared to past UV/VIS spaceborne sensors, reduces the representation error of each retrieved $NO_2$ and HCHO pixel (Souri et al., 2022b). In this study, we apply daily TROPOMI tropospheric column $NO_2$ v2.3.1 (van Geffen et al., 2022) and HCHO v1.1.5 retrievals (De Smedt et al., 2018). For TROPOMI $NO_2$ data we used the product provided by the Product Algorithm Laboratory

(PAL) which applies the $NO_2$ v2.3.1 algorithm but for the time period between April 2018 - September 2021. The retrievals of both species use QA4ECV methods described above applying the DOAS methods with spectral fitting windows between 405 and 465 nm for $NO_2$ (Boersma et al., 2018) and 328.5 and 346.0 nm for HCHO (De Smedt et al., 2018). TROPOMI retrievals are similar to those from the QA4ECV OMI product as it applies the same a priori profiles from the TM5 model, albedo data, and cloud fraction information. TROPOMI $NO_2$ v2.3.1 retrievals do differ

from QA4ECV OMI $NO_2$ products as it uses cloud pressure input from the $O_2$-A band following the FRESCO-wide approach (van Geffen et al., 2022) instead of $O_2$–$O_2$ absorption. Similarly, TROPOMI HCHO v1.1.5 retrievals differ from the QA4ECV OMI HCHO data through applying the S5P ROCINN algorithm which uses the $O_2$-A for cloud pressures (Loyola et al., 2018) instead of $O_2$–$O_2$ absorption.

## 2.3 TEMPO synthetic retrieval product

One component of the pre-launch activities of the geostationary TEMPO satellite mission is to generate synthetic retrieval data for end-user communities, which closely represents the planned operational products of the mission planned for launch in early 2023 (Naeger et al., 2021). The synthetic data products are provided daily and at the expected 2.0 km × 4.75 km (at nadir) spatial resolution of TEMPO. Synthetic TEMPO data is applied in this study to demonstrate the additional FNR information, which will be provided by the high spatiotemporal resolution (including



up to hourly information during the daylight hours) of this geostationary sensor, compared to existing low earth orbit sensors (i.e., OMI and TROPOMI).

Hourly model output of $NO_2$ and HCHO vertical profiles from the NASA Global Modeling and Assimilation Office (GMAO) GEOS Composition Forecasting (GEOS-CF) model were used as input into the TEMPO proxy development methodology and sampled at the TEMPO footprint (2.0 km × 4.75 km at the center of the Field of

Regard) to represent the "true" state of the atmosphere. GEOS-CF simulates meteorology and aerosol and trace gas concentrations in both the troposphere and stratosphere at a high global spatiotemporal resolution (0.25° × 0.25°) and 72 vertical layers (Knowland et al., 2020, 2022; Keller et al., 2020). The GEOS-CF model is a reliable source for the "true" atmosphere as it has been shown to produce realistic concentrations of aerosols and trace gases in comparison to remote-sensing observations and in situ measurements (e.g., Keller et al., 2020; Johnson et al., 2021; Knowland et

al., 2022). Scattering weights in clear and cloudy conditions are derived from pre-computed lookup tables of radiances and $NO_2$ and HCHO scattering weights at 440 and 340 nm, respectively, as a function of the TEMPO viewing geometry centered at 91°W, surface reflectance, cloud fraction, cloud pressure, and absorption by $O_3$. The Geostationary Coastal and Air Pollution Events (GEOCAPE) Radiative Transfer Tool (based on the Vector Linearized Discrete Ordinate Radiative Transfer (VLIDORT) model (Spurr, 2006)) was used to create the lookup table. A fast

optical centroid pressure simulator (Joiner et al., 2012), a simple mixed Lambertian model where clouds are parameterized as opaque reflective surfaces, was used to account for the effects of clouds on the SWs. Surface model reflectance is based on Moderate Resolution Imaging Spectroradiometer (MODIS) Blue Sky Albedo calculations, and the Cox-Munk Glitter kernel + whitecap parameterization + water leaving radiances. Climatology albedo is based on the OMI Lambertian equivalent reflectance. After deriving the SWs from the lookup table, "true" SCDs are calculated

from the summation of the target trace gas (i.e., $NO_2$, HCHO) concentration profile multiplied by the scattering weights in each model layer.

To convert the true SCD to the final proxy VCD, we i) applied a statistical random noise model developed in Zoogman et al. (2017) for TEMPO proxy data, primarily as a function of spectral signal-to-noise ratios and column abundance of $NO_2$ and HCHO, to the "true" SCDs, ii) applied climatological AMFs derived from $NO_2$ and HCHO scattering weights based on the same lookup table approach as discussed above, but using $NO_2$ and HCHO profiles

from hourly model output data from the Goddard Earth Observing System Model version 5 with GEOS-Chem as a chemical model at ~12 km grid spacing (G5NR-Chem; Hu et al., 2018).

### 2.4 Airborne spectrometers

The primary evaluation data set used in this study is from the UV/VIS airborne remote-sensing data product from

GeoTASO and GCAS flown during the LISTOS 2018 field campaign (16 flights between June 18 and October 19, 2018). Due to the fact that no bias-corrected tropospheric column HCHO data is available during LISTOS 2018 from the Pandora network, this ground-based remote-sensing network is not applied here. Both the GeoTASO and GCAS instruments and retrievals are very similar and together provide a consistent evaluation data set (see specific details on the instruments and $NO_2$ and HCHO retrievals in Kowalewski and Janz (2014), Leitch et al. (2014), Nowlan et al.

(2016, 2018), and Judd et al. (2020)). GeoTASO and GCAS $NO_2$ and HCHO data were obtained from a nominal flight



altitude of 9 km above ground level (agl) covering the majority of the troposphere. The airborne data from 13 flight days between June 25 and September 6, 2018 (see Table 1) are provided with a native spatial resolution of 250 m × 250 m. To reduce noise in the raw GeoTASO and GCAS retrievals, the data were averaged to a 1 km × 1 km spatial resolution. In total, measurements from 8 and 12 flight days were spatiotemporally co-located with OMI and

TROPOMI overpasses, respectively. A detailed explanation of the measurements and flights conducted during LISTOS 2018 is provided in Judd et al. (2020).

      The airborne GeoTASO and GCAS retrievals are used here as the reference data set for validating all satellite data. However, the airborne remote-sensing data is not without error. A nearly identical GeoTASO and GCAS tropospheric column $NO_2$ data set used in this work was applied in Judd et al. (2020) and was evaluated with a network

of Pandora systems. Judd et al. (2020) demonstrated that the airborne $NO_2$ retrievals had a median bias of ~1% with uncertainty within ±25% with no magnitude dependent biases. Due to minimal availability of ground-based remote-sensing Pandora data of HCHO, airborne GeoTASO and GCAS retrievals of this species has had limited evaluation. Nowlan et al. (2018) did evaluate GCAS tropospheric HCHO retrievals using P-3B airborne in situ measurements and determined GCAS had generally good performance with a < 10% bias (minimal magnitude dependance in bias) and

high correlation with observations. Overall, the satisfactory comparison of airborne GeoTASO and GCAS tropospheric column $NO_2$ and HCHO with independent observations provides confidence that this data can be applied as a reference data set to validate OMI and TROPOMI retrievals. However, it should be kept in mind that there is some level of error/bias associated with the GeoTASO and GCAS data used in this study (e.g., Nowlan et al., 2016; 2018; Judd et al., 2020).

**Table 1. Airborne (GeoTASO and GCAS) flight information (date, flight times, number of co-located satellite and airborne FNR grids) used in this study from the LISTOS 2018 field campaign.**

| Flight Day Number | Date | Time (Hours in UTC) | OMI FNR co-locations[1] | TROPOMI FNR co-locations[2] |
|---|---|---|---|---|
| 1 | June 25, 2018 | Morning: 12.5–15.7 <br> Afternoon: 16.8–20.3 | 12 | 201 |
| 2 | June 30, 2018 | Morning: 12.2–15.6 <br> Afternoon: 16.7–20.4 | 37 | 251 |
| 3 | July 2, 2018 | Morning: 11.4–16.6 <br> Afternoon: 17.9–21.5 | 6 | 66 |
| 4 | July 19, 2018 | Morning: 11.4–15.3 <br> Afternoon: 16.9–20.9 | 0 | 155 |
| 5 | July 20, 2018 | Morning: 11.4–15.3 <br> Afternoon: 17.1–21.1 | 5 | 136 |
| 6 | August 5, 2018 | Morning: 12.5–16.5 <br> Afternoon: 17.8–22.3 | 5 | 0 |
| 7 | August 6, 2018 | Morning: 11.7–16.0 <br> Afternoon: 17.2–21.5 | 0 | 67 |
| 8 | August 15, 2018 | Morning: 11.2–15.5 <br> Afternoon: 17.0–21.6 | 0 | 150 |
| 9 | August 16, 2018 | Morning: 11.3–15.3 | 0 | 108 |





| | | | OMI FNR[1] | TROPOMI FNR[2] |
|---|---|---|---|---|
| | | Afternoon: 17.3–21.5 | | |
| 10 | August 24, 2018 | Morning: 10.9–15.3 | 20 | 147 |
| | | Afternoon: 16.6–21.0 | | |
| 11 | August 28, 2018 | Morning: 11.3–15.3 | 8 | 150 |
| | | Afternoon: 16.6–20.3 | | |
| 12 | August 29, 2018 | Morning: 11.2–15.1 | 0 | 166 |
| | | Afternoon: 16.6–20.8 | | |
| 13 | September 6, 2018 | Morning: 11.9–15.8 | 8 | 96 |
| | | Afternoon: 17.2–21.4 | | |

[1]OMI FNR co-locations for the near-native 0.15° × 0.15° spatial resolution gridded data.
[2]TROPOMI FNR co-locations for the near-native 0.05° × 0.05° spatial resolution gridded data.

### 2.5 CMAQ model simulation

The prior vertical profiles play a major role in satellite retrievals of chemical constituents in the troposphere (e.g., Palmer et al., 2001; Boersma et al., 2007; Johnson et al., 2018). Furthermore, past research has demonstrated that using a well-constrained, high spatial resolution, air quality model or CTM as the a priori profile source for satellite retrievals can improve VCD results (e.g., Laughner et al., 2019). To compare NASA OMI and TROPOMI tropospheric $NO_2$, HCHO, and FNR retrievals using a common a priori profile data set, we conduct sensitivity tests applying model

simulated vertical profiles of $NO_2$ and HCHO produced by the Community Multiscale Air Quality Model (CMAQ). Reprocessing OMI and TROPOMI $NO_2$ and HCHO retrievals with a common, high spatial resolution ($4 \times 4$ km$^2$), model data product removes differences in the satellite products due to using different coarse spatial resolution model data sources as a priori vertical profiles.

We used CMAQ version 5.3 for air quality simulations during the LISTOS 2018 campaign. The CMAQ

simulations were driven offline using the meteorological fields simulated by the Weather Research and Forecasting (WRF) model version 4.1. The WRF-CMAQ spatial domain set-up is shown in Fig. S1. The outer WRF domain covers the contiguous United States (CONUS) at a horizontal grid spacing of $12 \times 12$ km$^2$ ($481 \times 369$ grid points) and the inner WRF domain covers the northeastern US, encompassing the entire LISTOS 2018 campaign domain, at a horizontal grid spacing of $4 \times 4$ km$^2$ ($237 \times 189$ grid points). Both the outer and inner model domains use 35 vertical

levels between the surface and 50 hPa. The WRF configuration follows Appel et al. (2017), which includes improved representation of the land-surface processes and vertical mixing, and employs four-dimensional data assimilation (also called grid nudging) every 6 hours to limit the growth of meteorological errors in the simulations (WRF configuration details in Table S1). A 15-day spin up period was used for the WRF-CMAQ simulations to minimize the impacts of errors in initial conditions. Anthropogenic emissions of trace gases and aerosols are based on the National Emissions

Inventory (NEI) representative of 2014 because that was the latest available inventory from EPA at the time of emission preparation. NEI 2014 emissions were processed using the Sparse Matrix Operator Kernel Emissions (SMOKE) model with the same configuration as adopted in the EPA 2014 emissions modeling platform (https://www.epa.gov/air-emissions-modeling/2014-version-71-platform). The same WRF simulations described above were used to drive SMOKE for generating meteorology-dependent anthropogenic emissions. Biogenic

emissions of trace gases and aerosols are calculated online within the model using the Biogenic Emissions Inventory



System (BEIS). The gas-phase chemistry and aerosol processes are represented using Carbon bond 6 (CB06) version r3 with AERO7 treatment of the secondary organic aerosols. Chemical lateral boundary conditions for the outer domain were based on the idealized profiles available in CMAQ but are dynamically provided to the inner domain every hour based on the outer domain simulations.

**2.6 Evaluation techniques**

In order to perform a systematic, direct comparison of daily satellite products to airborne retrievals, OMI and GeoTASO/GCAS data were spatially-averaged to $0.15° \times 0.15°$ (~$15 \times 15$ km$^2$, similar to OMI nadir spatial resolution) for evaluating OMI. TROPOMI and airborne observations were spatially-averaged at $0.05° \times 0.05°$ (~$5 \times 5$ km$^2$, similar to TROPOMI nadir spatial resolution) for evaluating TROPOMI data. To investigate the impact of the higher spatial resolution of TROPOMI, NO$_2$, HCHO, and FNR retrievals from this sensor were also averaged to the $0.15° \times 0.15°$ for inter-comparison with OMI evaluation statistics. In order to smooth and reduce the noise of satellite data, we apply a point oversampling technique (e.g., McLinden et al., 2012) when spatially averaging the retrievals. This method uses a larger grid box radius, compared to the averaging resolution, to bin individual retrievals. When averaging satellite data to the $0.15° \times 0.15°$ spatial resolution (standard radius of 0.075°), we employed a radius twice the standard size equal to 0.15°. Similarly, when averaging satellite data to the $0.05° \times 0.05°$ spatial resolution (standard radius of 0.025°) we applied a radius of 0.05°. By spatially-averaging the tropospheric column NO$_2$ and HCHO GeoTASO/GCAS data we minimized the spatial representation error between OMI and TROPOMI satellite retrieved pixels with those of GeoTASO/GCAS.

Given that the nominal flight altitude for GeoTASO and GCAS observations was 9 km agl, in order to directly compare to satellite tropospheric column retrievals, we scaled airborne tropospheric column NO$_2$ values by multiplying the observed values by the ratio of the total tropospheric NO$_2$ column abundance over the tropospheric column NO$_2$ abundance below 9 km agl (i.e., $\frac{\int Tropospheric\ NO_2\ (surface\ to\ tropopause)}{\int Tropospheric\ NO_2\ (surface\ to\ 9\ km\ agl)}$). This scaling factor for NO$_2$, which showed that typically 60% to 99% of tropospheric NO$_2$ is below 9 km agl, was derived for each co-located GeoTASO and GCAS retrieval, using the WRF-CMAQ simulations described in Sect. 2.5. Tropospheric column HCHO data from GeoTASO and GCAS were not scaled due to the fact that typically >95% of the total column HCHO is below the nominal aircraft flight altitude.

For comparison to satellite retrievals, GeoTASO and GCAS data were co-located to OMI and TROPOMI data using a temporal threshold of $\pm$ 60 minutes. Before GeoTASO and GCAS HCHO and NO$_2$ data were co-located with satellite data they were filtered to remove airborne retrievals where the radiance flag was > 0.5 as they are considered to be influenced by clouds or glint. We initially applied a temporal threshold of $\pm$ 30 minutes; however, this resulted in < 50 total co-locations with OMI retrievals throughout the study time period. Therefore, the longer temporal threshold criteria was necessary to achieve enough co-locations for statistical evaluation. The longer temporal threshold of $\pm$ 60 minutes resulted in only slightly larger median biases compared to when applying the $\pm$ 30 minute threshold. The similar bias statistics using temporal offsets of 30 and 60 minutes agrees with other studies which show minimal dependance on temporal offsets between 0 and 60 minutes (e.g., Tack et al., 2021). It should be noted that the temporal threshold of $\pm$ 60 minutes, and spatial gridding/averaging methods applied in this study,



resulted in slightly larger spread in TROPOMI $NO_2$ data when evaluated to GeoTASO and GCAS data compared to the results in Judd et al. (2020) which used a ± 30 minute co-location threshold.

Satellite retrievals with high quality were filtered for use by removing individual retrievals that did not have quality flags (qa) = 0 for HCHO and $NO_2$ when applying OMI data. For TROPOMI, individual retrievals of $NO_2$ and HCHO that had qa < 0.75 and qa < 0.5 were removed prior to spatial averaging, respectively. These qa values were selected based on guidance from OMI and TROPOMI user's guides to remove data with large uncertainty to produce high quality science data products. Furthermore, to avoid anomalous OMI and TROPOMI retrieval values of HCHO, we remove VCDs with lower and upper bounds of $-8.0 \times 10^{15}$ and $7.6 \times 10^{16}$ molecules cm$^{-2}$, respectively. These

bounds were determined from typical HCHO VCD values and a threshold of 3 times the fitting uncertainty of OMI retrievals following Zhu et al. (2020). Similarly, to avoid anomalous OMI and TROPOMI retrieval values of $NO_2$, we remove VCDs with lower and upper bounds of $-1.08 \times 10^{15}$ and $8.07 \times 10^{16}$ molecules cm$^{-2}$, respectively (personal communication with OMI $NO_2$ algorithm team). Both OMI and TROPOMI retrievals with solar zenith angles > 70° and effective cloud fractions > 30% and > 50%, respectively were also removed. These additional thresholds were

chosen based on guidance from the OMI and TROPOMI user's guides. Finally, only co-located spatially-averaged grids that had 75% spatial coverage by GeoTASO/GCAS data and airborne remote-sensing $NO_2$ VCDs > $1.0 \times 10^{15}$ molecules cm$^{-2}$ were used in the evaluation.

The statistical evaluation of daily and campaign-averaged (includes all flights displayed in Table 1) OMI and TROPOMI retrievals with co-located GeoTASO and GCAS spatially-averaged data was primarily done using bias

(median), oscillation/variability in bias represented by the standard deviation of bias (referred to as bias standard deviation throughout), normalized median bias (NMB) which are normalized to the magnitude of observed data, and simple linear regression statistics (slope, y-intercept, coefficient of determination ($R^2$)) based on ordinary least-squares.

## 3 Results

In this section we evaluate the capability of NASA OMI products (hereinafter referred to as NASA OMI), QA4ECV OMI retrievals (hereinafter referred to as QA4ECV), and TROPOMI to retrieve tropospheric columns of $NO_2$, HCHO, and FNRs during the LISTOS 2018 (temporally-averaged values from all flights hereinafter referred to as campaign-averaged). We further evaluate these retrievals on a day characterized by large $NO_2$ pollution focusing on NASA OMI and TROPOMI. We also present results of a sensitivity test using common a priori vertical profiles of $NO_2$ and HCHO

from WRF-CMAQ to reprocess NASA OMI and TROPOMI retrievals. Finally, we present information on the expected additional FNR information that will be provided from the future NASA geostationary TEMPO satellite.

### 3.1 Campaign-averaged tropospheric FNRs

Airborne observations during the summer of 2018 suggest that during the mid-day hours large regions of FNRs ≤ 1.0 occurred over the urban regions surrounding New York City (NYC). The term "urban" here is used qualitatively as

the region close in proximity to the center of NYC where elevated tropospheric column $NO_2$ values over $NO_x$ emission regions are frequently observed. The opposite is true for the usage of "rural" hereinafter. Figure 1 shows the campaign-



averaged FNRs from OMI (NASA and QA4ECV) and TROPOMI retrievals averaged to spatial resolutions of 0.15°
× 0.15° and 0.05° × 0.05°, respectively compared to co-located airborne remote-sensing products. These regions of
FNRs ≤ 1.0 likely have $O_3$ production which is limited by VOC emissions. Outside of the VOC/radical-limited region

around NYC, airborne observations show a clear transition zone of FNRs between 1.0 and 2.0 and $NO_x$-limited
regimes (FNR > 2.0) in the rural regions of the northeast US. It should be noted these FNR thresholds being discussed
follow the assumptions of Duncan et al. (2010); however, there are uncertainties in the exact thresholds separating $O_3$
sensitivity production regimes and they can be spatiotemporally variable (e.g., Lu and Chang, 1998; Schroeder et al.,
2017; Souri et al., 2020; Ren and Xie, 2022). For example, a recent study by Jin et al. (2020) suggests that

VOC/radical-limited regimes around NYC transition to $NO_x$-limited regimes for FNRs between 2.9 and 3.8. For
simplicity, we use the constant FNR ratio thresholds defined by Duncan et al. (2010) for discussion throughout the
rest of this study.

Satellite retrievals during the summer of 2018 also displayed the same general regional patterns of FNRs in
the northeast US that were observed by airborne remote-sensing (see Fig. 1). NASA OMI, QA4ECV, and TROPOMI

retrieved lower FNRs in the urban region of NYC and a transition to $NO_x$-limited regimes in the rural regions.
However, all satellite products show higher FNRs (between 1.0 and 3.0) in the areas where airborne observations
clearly observed $NO_x$-saturated regimes. In general, TROPOMI FNRs at the 0.05° × 0.05° spatial resolution have the
lowest values over NYC in better agreement with airborne observations. The higher spatial resolution satellite data
provided by TROPOMI also has a smaller spatial extent of a transition zone and VOC/radical-limited regimes in

comparison to the two OMI products. TROPOMI FNR retrievals and airborne observations display a clear urban/rural
interface; however, OMI products result in noisier spatial patterns. Between the two OMI retrieval products, QA4ECV
FNR values are lower in the observed VOC/radical-limited region in comparison to NASA OMI and appear to
compare more favorably to airborne observations.

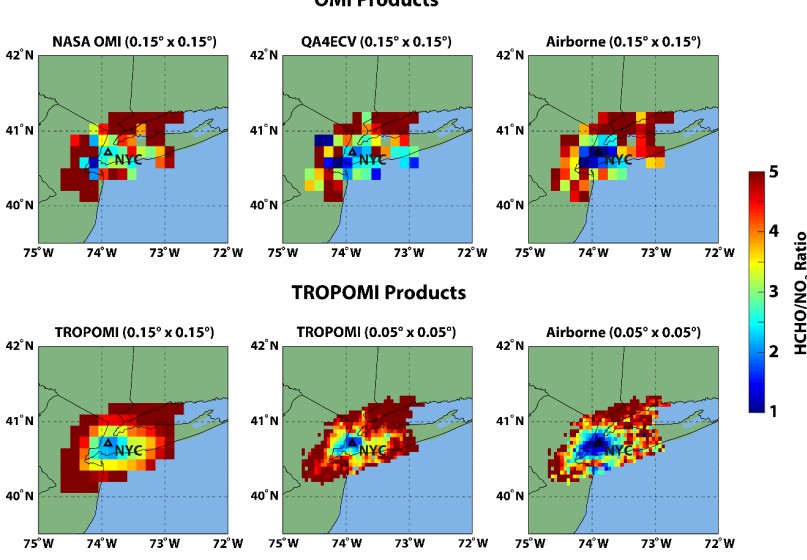



**Figure 1: NASA OMI, QA4ECV, TROPOMI, and airborne tropospheric column FNR retrievals averaged for all flights conducted during the LISTOS 2018 field campaign. All co-located OMI and airborne remote-sensing tropospheric column FNR values are averaged at $0.15° \times 0.15°$ and TROPOMI co-locations are averaged at both $0.05° \times 0.05°$ and $0.15° \times 0.15°$ spatial resolutions. The black triangle indicates the location of the city of NYC.**

Figure 1 illustrates the impact of retrieval spatial resolution on the ability of satellite-derived FNRs to
reproduce observed $O_3$ sensitivity production regimes. TROPOMI retrieval data better captures the spatial pattern and urban/rural interface of observed $O_3$ sensitivity production regimes compared to OMI data. TROPOMI results when gridded near the native resolution of the sensor ($0.05° \times 0.05°$), while still higher compared to observed FNRs around NYC, were able to retrieve FNRs < 2.0. However, when averaged to a resolution similar to the native resolution of OMI ($0.15° \times 0.15°$), TROPOMI data suggests higher FNRs ≥ 2.0 in the vicinity of NYC, in line with OMI retrieval
products.

It should be noted that satellite- and airborne-retrieved FNRs are dependent on both tropospheric $NO_2$ and HCHO values. Median/mean and unresolved biases in FNRs can then be driven by errors in either retrievals of $NO_2$ and/or HCHO. Therefore, the following sections of this work investigate the statistical comparison of NASA OMI, QA4ECV, and TROPOMI tropospheric $NO_2$, HCHO, and resulting FNRs compared to airborne observations.

**3.2 Statistical evaluation of OMI and TROPOMI retrievals**

**3.2.1 Tropospheric column NO₂ retrievals**

The spatial pattern of campaign-averaged tropospheric column $NO_2$ retrieved by the satellites and airborne sensors highlight the large pollution region around the urban region of NYC during the summer of 2018 (see Fig. S2). Tropospheric column $NO_2$ concentrations over NYC from both satellite and airborne observations frequently exceed
$1.0 \times 10^{16}$ molecules cm$^{-2}$ within 60 minutes of the OMI and TROPOMI overpass times. However, while airborne tropospheric column $NO_2$ values in the rural regions surrounding NYC were frequently observed to be < $2.0 \times 10^{15}$ molecules cm$^{-2}$, satellite retrievals have larger background tropospheric column $NO_2$ concentrations between $2.0 \times 10^{15}$ and > $4.0 \times 10^{15}$ molecules cm$^{-2}$. This suggests OMI and TROPOMI retrievals have a high bias in background tropospheric column $NO_2$ concentrations. This high bias in satellite background tropospheric column $NO_2$ values can
possibly be linked to underestimated abundance of free tropospheric $NO_2$ in CTMs used as a priori profile data sets for OMI and TROPOMI retrievals resulting in AMFs which are too low (e.g., Silvern et al., 2019). Furthermore, studies have shown that the coarse spatial resolution of the CTMs used to derive a priori $NO_2$ profiles for OMI and TROPOMI cannot resolve the sharp gradients of $NO_2$ at the urban/rural interface and lead to the overestimate of satellite retrievals in low pollution regions (Lamsal et al., 2014; Tack et al., 2021). Finally, other aspects of the satellite
retrievals such as biases in stratospheric $NO_2$ concentrations and separation from the troposphere, aerosol interference, and surface albedo could contribute to these overestimations in background, low pollution regions (e.g., Lamsal et al., 2021).

Figure 2 shows the comparison of co-located NASA OMI, QA4ECV, and TROPOMI retrievals of tropospheric column $NO_2$ concentrations with observed data from all flights (statistical evaluation shown in Table 2).
This figure and Table 2 further emphasize the high bias of background tropospheric column $NO_2$ concentrations retrieved by NASA OMI, QA4ECV, and TROPOMI. All satellite products typically have a high bias compared to the



small tropospheric column $NO_2$ concentrations ($< 5.0 \times 10^{15}$ molecules $cm^{-2}$) measured outside the urban regions of NYC, linear regression slopes $< 0.65$, and positive y-intercepts when compared to the airborne observations. Some of this high bias in background tropospheric column $NO_2$ concentrations is offset in the campaign-averaged median

biases by the fact that the satellite retrievals have a low bias compared to $NO_2$ values observed over polluted regions ($> 1.0 \times 10^{16}$ molecules $cm^{-2}$).

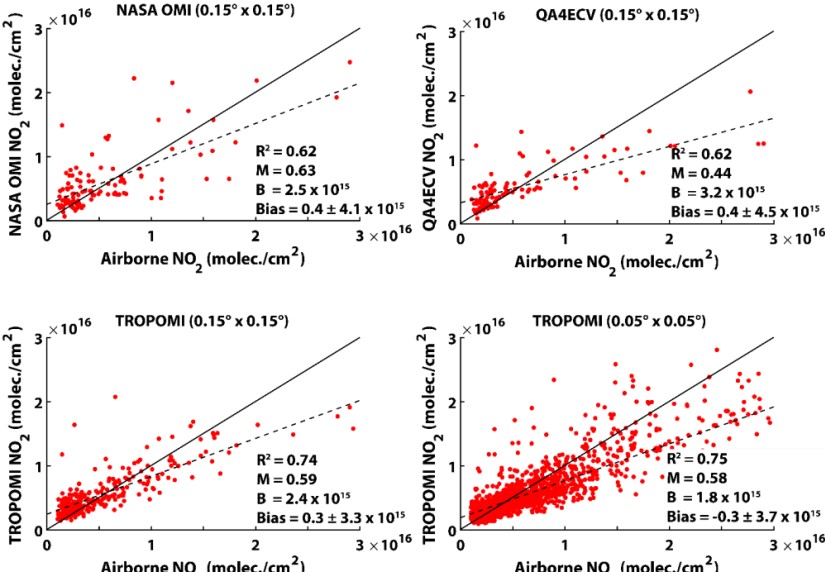

**Figure 2: Scatter plots illustrating the comparison of satellite- (NASA OMI, QA4ECV, and TROPOMI) and airborne-retrieved tropospheric $NO_2$ (molecule $cm^{-2}$) for each co-located measurement taken during the field campaign. All co-**
**located OMI and airborne remote-sensing tropospheric column $NO_2$ values are averaged at the $0.15° \times 0.15°$ resolution and TROPOMI co-located data are averaged at $0.15° \times 0.15°$ and $0.05° \times 0.05°$ spatial resolution. The solid black line shows the 1:1 comparison and the dashed line shows the linear regression fit of the comparison. The figure inset shows the main statistics (coefficient of determination ($R^2$), slope (M), y-intercept (B), and median bias and bias standard deviation) of the comparison of satellite and airborne tropospheric column $NO_2$ data.**

475        NASA OMI displays a small campaign-averaged median bias (NMB %) of $0.4\pm4.1 \times 10^{15}$ molecules $cm^{-2}$ (6.3%) in comparison to tropospheric column $NO_2$ observations. QA4ECV OMI data results in a campaign-averaged median bias of $0.4\pm4.5 \times 10^{15}$ molecules $cm^{-2}$ (6.8%). Finally, TROPOMI retrievals have a campaign-averaged median bias of $-0.3\pm3.7 \times 10^{15}$ molecules $cm^{-2}$ (-4.8%) and $0.3\pm3.3 \times 10^{15}$ molecules $cm^{-2}$ (5.8%) when averaged at $0.05° \times 0.05°$ and $0.15° \times 0.15°$ spatial resolution, respectively. It should be noted that the TROPOMI low bias in tropospheric

column $NO_2$ is improved with the newer retrieval algorithm used in this study compared to early versions of the data product (e.g., v1.2.2 had a campaign-averaged median low bias of $-1.3\pm4.0 \times 10^{15}$ molecules $cm^{-2}$). In addition to mean/median biases, bias standard deviation, which is indicative of noise in the satellite retrievals, is very important for accurate retrievals of the spatial-resolved daily tropospheric column $NO_2$, HCHO, and FNRs. At the near-native spatial resolution of the three satellite retrievals, the standard deviation in bias of the data were similar ($\sim4.0 \times 10^{15}$

molecules $cm^{-2}$) with QA4ECV OMI data having the largest bias standard deviation and TROPOMI having the least





noise in the data. TROPOMI data averaged to match OMI spatial resolution displayed lower bias standard deviation values of ~$3.0 \times 10^{15}$ molecules cm$^{-2}$. At both spatial resolutions, TROPOMI tropospheric NO$_2$ data has slightly less spread compared to OMI products. The results here suggest that OMI and TROPOMI tropospheric column NO$_2$ retrievals errors have a magnitude dependence and tend to have a high bias in rural/background regions and a low bias

in moderately to highly polluted regions which agrees with past validation studies (e.g., Judd et al., 2020; Compernolle et al., 2020; Lamsal et al., 2021).

      To determine if the higher spatial resolution of TROPOMI resulted in more favorable comparisons to observations, we compare TROPOMI tropospheric column NO$_2$ values to OMI results. TROPOMI tropospheric column NO$_2$ concentrations at 0.05° × 0.05° displayed the lowest campaign-averaged median bias of all satellite

products, and the higher spatial resolution data better reproduces the spatial patterns of observed tropospheric column NO$_2$. This is emphasized by the higher correlation when evaluating TROPOMI tropospheric column NO$_2$ concentrations with observations in comparison to the other satellite products and visually more clearly separating the urban/rural interface seen in tropospheric NO$_2$ (see Fig. S2).

**Table 2. Statistical evaluation of NASA OMI, QA4ECV, and TROPOMI retrievals of tropospheric column NO$_2$ and HCHO and resulting FNRs. Statistics presented are the number of co-located grids (N), median bias ± bias standard deviation, NMB (%), coefficient of determination (R$^2$), and linear regression slope (Slope).**

| NASA OMI (0.15° × 0.15°) | | | | QA4ECV (0.15° × 0.15°) | | |
|---|---|---|---|---|---|---|
| | FNR | HCHO* | NO$_2$* | | FNR | HCHO* | NO$_2$* |
| N | 101 | 101 | 116 | N | 82 | 85 | 106 |
| Bias | 0.4±3.8 | 5.1±7.8 | 0.4±4.1 | Bias | -0.2±3.3 | 2.3±8.9 | 0.4±4.5 |
| NMB | 11.0 | 38.7 | 6.3 | NMB | -5.4 | 17.3 | 6.8 |
| R$^2$ | 0.23 | 0.19 | 0.62 | R$^2$ | 0.17 | 0.19 | 0.62 |
| Slope | 1.0 | 0.46 | 0.63 | Slope | 0.67 | 0.54 | 0.44 |
| TROPOMI (0.15° × 0.15°) | | | | TROPOMI (0.05° × 0.05°) | | |
| | FNR | HCHO* | NO$_2$* | | FNR | HCHO* | NO$_2$* |
| N | 261 | 261 | 261 | N | 1693 | 1741 | 1802 |
| Bias | 0.3±1.4 | 2.9±4.9 | 0.3±3.3 | Bias | 0.4±2.3 | 1.9±6.7 | -0.3±3.7 |
| NMB | 9.3 | 23.1 | 5.8 | NMB | 13.0 | 12.9 | -4.8 |
| R$^2$ | 0.48 | 0.40 | 0.74 | R$^2$ | 0.29 | 0.28 | 0.75 |
| Slope | 0.75 | 0.47 | 0.59 | Slope | 0.70 | 0.55 | 0.58 |

*bias units are ×$10^{15}$ molecules cm$^{-2}$.

### 3.2.2 Tropospheric column HCHO retrievals

The spatial pattern of campaign-averaged tropospheric column HCHO retrieved by the satellites and airborne sensors

highlight the large HCHO concentrations in both urban and rural regions during the summer of 2018 (see Fig. S3). This differs from tropospheric column NO$_2$, which is primarily emitted from anthropogenic sources, due to the fact HCHO has both anthropogenic and natural precursor emission sources and precursors with longer atmospheric lifetime. The longer lifetime of precursor species producing HCHO result in less heterogeneity and gradients in HCHO



concentrations throughout the domain. Airborne observations of tropospheric column HCHO concentrations show

that over NYC the concentrations are on average ~$1.5 \times 10^{16}$ molecules cm$^{-2}$, and can exceed $2.5 \times 10^{16}$ molecules cm$^{-2}$ during the afternoon hours (see Fig. S3). Both OMI and TROPOMI retrieval products have smaller gradients between HCHO concentrations in the urban and rural regions in comparison to airborne observations.

Figure 3 shows the scatter plot comparison of co-located NASA OMI, QA4ECV, and TROPOMI retrievals of tropospheric column HCHO concentrations compared to observed data (statistical evaluation shown in Table 2).

This figure and Table 2 illustrate the high bias of background tropospheric column HCHO concentrations retrieved by NASA OMI, QA4ECV, and TROPOMI compared to airborne observations. All satellite products have a high bias when tropospheric columns HCHO are $\leq 1.5 \times 10^{16}$ molecules cm$^{-2}$, linear regression slopes < 0.60, and positive y-intercepts when compared to observations (in agreement with Vigouroux et al. (2020)). Both OMI retrieval products and TROPOMI data better replicate the larger HCHO concentrations (between $1.5 \times 10^{16}$ and $3.0 \times 10^{16}$ molecules

cm$^{-2}$) with some small low bias in more polluted regions (> $3.0 \times 10^{16}$ molecules cm$^{-2}$). On average, NASA OMI had the largest campaign-averaged median high bias of $5.1\pm7.8 \times 10^{15}$ molecules cm$^{-2}$ (38.7%). QA4ECV OMI data results in a lower campaign-averaged median high bias of $2.3\pm8.9 \times 10^{15}$ molecules cm$^{-2}$ (17.3%). Finally, TROPOMI retrievals had the lowest campaign-averaged median high bias of $1.9\pm6.7 \times 10^{15}$ molecules cm$^{-2}$ (12.9%) at 0.05° × 0.05° spatial resolution and $2.9\pm4.9 \times 10^{15}$ molecules cm$^{-2}$ (23.1%) when averaged at 0.15° × 0.15°. Spatially

averaging TROPOMI tropospheric column HCHO, along with tropospheric column NO$_2$ and FNRs, to coarser grids in order to increase signal-to-noise aided in reducing the bias standard deviation in HCHO retrieval products (see Table 2).

The results of the validation shown in Fig. 3 and Table 2 are consistent with recent validation studies such as the work of Vigouroux et al. (2020) and De Smedt et al. (2021) which also show that in regions of high tropospheric

column HCHO concentrations, OMI and TROPOMI retrievals are generally consistent with some moderate low bias. However, in regions of lower background tropospheric column HCHO concentrations, both OMI and TROPOMI HCHO retrievals are biased high and OMI products tend to display a larger high bias compared to TROPOMI. Furthermore, these two studies agree with our analysis that TROPOMI HCHO has lower bias standard deviation, and higher correlations with observations, compared to both OMI products evaluated here. The larger spread in

tropospheric HCHO from OMI compared to TROPOMI is likely due to the weaker signal-to-noise in OMI and potentially the fewer co-located data points for statistical analysis. This is further demonstrated by the TROPOMI bias standard deviation being nearly a factor of two smaller compared to NASA OMI and QA4ECV when averaged to the OMI spatial resolution. TROPOMI HCHO retrievals have the smallest median bias and bias standard deviation compared to observations, and highest correlation with airborne observations, suggesting this newer sensor can better

retrieve HCHO compared to OMI during this time period.

All three satellite HCHO products have larger bias standard deviations and low correlations, when compared to the statistical evaluation of satellite NO$_2$ retrievals, when evaluated with observed tropospheric HCHO data. This highlights the large noise in these retrieval products likely driven by low signal-to-noise in HCHO retrievals. Furthermore, UV/VIS retrievals at shorter wavelengths (~340 nm) have much smaller sensitivity to HCHO compared

to longer wavelengths (~440 nm) employed for NO$_2$ retrievals (Lorente et al., 2017). The sensitivity of UV/VIS





retrievals to HCHO is lower throughout the middle and lower troposphere compared to $NO_2$, due to stronger Rayleigh scattering at shorter wavelengths, approaching twice as low near the surface (Lorente et al., 2017). The higher sensitivity of $NO_2$ retrievals in the middle to lower troposphere, compared to HCHO, is important as the highest concentrations, and largest spatiotemporal variability, of both $NO_2$ and HCHO occur lower in the troposphere near the

PBL likely leading to the higher correlation and lower bias standard deviation in the tropospheric column $NO_2$ statistical evaluation.

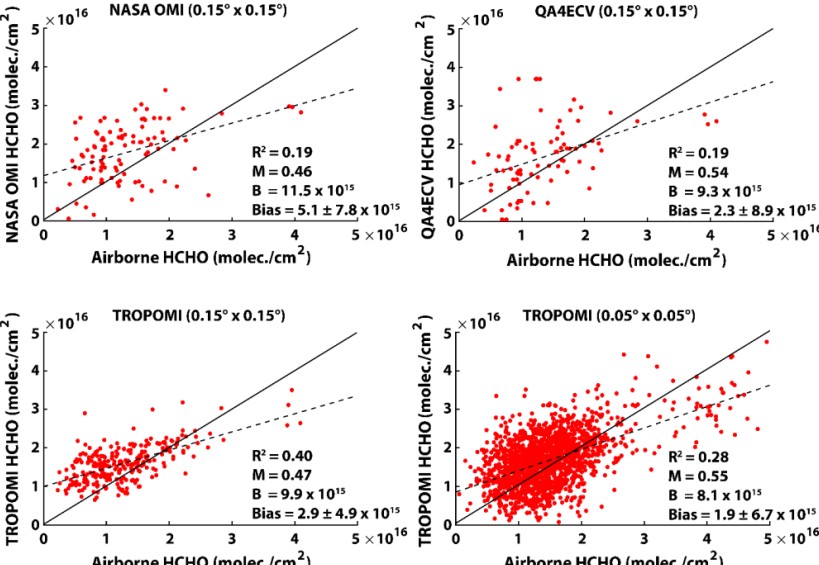

**Figure 3: Scatter plots illustrating the comparison of satellite- (NASA OMI, QA4ECV, and TROPOMI) and airborne-retrieved tropospheric HCHO (molecule cm$^{-2}$) for each co-located measurement taken during the field campaign. All co-located OMI and airborne remote-sensing tropospheric column HCHO values are averaged at the 0.15° × 0.15° resolution and TROPOMI co-located data are averaged at 0.15° × 0.15° and 0.05° × 0.05° spatial resolution. The solid black line shows the 1:1 comparison and the dashed line shows the linear regression fit of the comparison. The figure inset shows the main statistics (coefficient of determination ($R^2$), slope (M), y-intercept (B), and median bias and bias standard deviation) of the comparison of satellite and airborne tropospheric column HCHO data.**

**3.2.3 Tropospheric column FNR retrievals**

The spatial distribution of tropospheric FNRs observed by aircraft measurements during LISTOS 2018 was discussed previously (see Sect. 3.1). Here we evaluated the accuracy of NASA OMI, QA4ECV, and TROPOMI retrieved FNRs compared to observations. Figure 4 shows the scatter plot comparison of co-located NASA OMI, QA4ECV, and TROPOMI retrievals of tropospheric column FNRs compared to observed data (statistical evaluation shown in Table

2). NASA OMI displays a campaign-averaged median bias of 0.4±3.8 (11.0%) and QA4ECV OMI data resulted in a campaign-averaged median bias of -0.2±3.3 (-5.4%). TROPOMI retrievals had a campaign-averaged median bias of 0.4±2.3 (13.0%) and 0.3±1.4 (9.3%) when averaged at 0.05° × 0.05° and 0.15° × 0.15° spatial resolution, respectively. NASA OMI, QA4ECV, and TROPOMI FNR retrievals had similar biases compared to observations when averaged at coarser spatial resolutions (see Table 2). Regardless of how tropospheric column $NO_2$ and HCHO compared to







observations, all satellite products evaluated here resulted in campaign-averaged median biases ≤ 0.4 suggesting that
the mean of biases in the individual proxy species can offset to result in accurate FNR values.

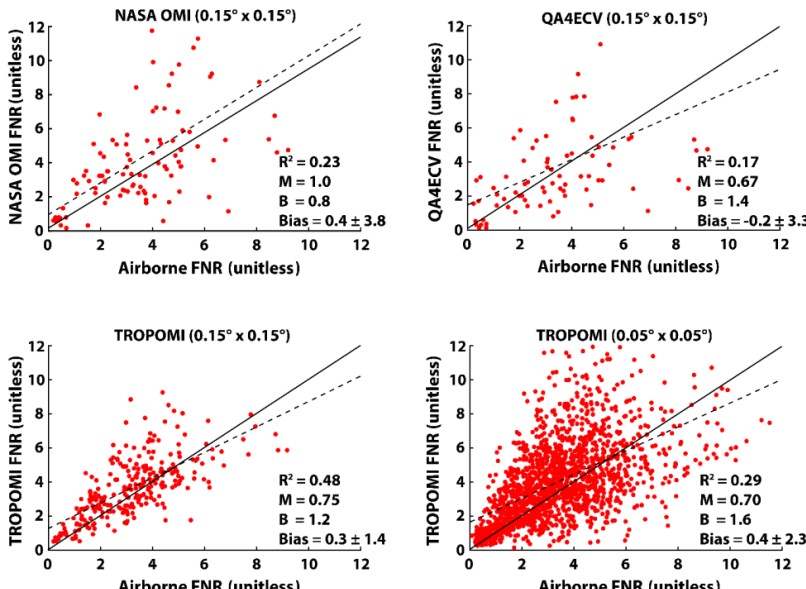

**Figure 4: Scatter plots illustrating the comparison of satellite- (NASA OMI, QA4ECV, and TROPOMI) and airborne-retrieved tropospheric FNR (unitless) for each co-located measurement taken during the field campaign. All co-located**

**OMI and airborne remote-sensing tropospheric column FNR values are averaged at the 0.15° × 0.15° resolution and TROPOMI co-located data are averaged at 0.15° × 0.15° and 0.05° × 0.05° spatial resolution. The solid black line shows the 1:1 comparison and the dashed line shows the linear regression slope of the comparison. The figure inset shows the main statistics (coefficient of determination ($R^2$), slope (M), y-intercept (B), and median bias and bias standard deviation) of the comparison of satellite and airborne tropospheric column FNR data.**

Visual inspection of TROPOMI and QA4ECV OMI retrievals suggests that these two products have the best ability to replicate the lowest observed FNRs over NYC during the field campaign (see Fig. 1). However, besides NASA OMI retrievals, the satellite products have linear regression slopes < 1.0 indicating a high bias for lower FNR values and some small low bias for higher observed FNRs. NASA OMI had a constant offset (slope = 1.0) of 0.8 for all values of observed FNRs.

The results of this study emphasize that the ability of satellites to accurately observe spatiotemporal patterns of daily FNRs is dependent on retrievals of both tropospheric column HCHO and $NO_2$. All three satellite products displayed high correlation with tropospheric column $NO_2$ observations, suggesting these spaceborne sensors can accurately assess the spatial patterns of this species. However, all the satellite products had very low correlation with observations of tropospheric HCHO, directly resulting in the low correlation of satellite FNR values compared to

observations. In fact, the rank in correlation levels of all four FNR satellite products evaluated here directly matches the rank in correlation levels of tropospheric HCHO. This leads to the conclusion that given bias variability in satellite tropospheric HCHO are large due to noise in the retrieval and low measurement sensitivity of shorter wavelengths in the troposphere, and directly drives the bias variability in FNR retrievals, satellite HCHO observations are the limiting





factor of using spaceborne retrievals to accurately assess daily FNRs for investigating $O_3$ chemistry and sensitivity
       regimes. It should be noted that the HCHO validation data from GeoTASO and GCAS are also hindered by weak
       absorption signatures in the shorter UV/VIS wavelengths and could add to the bias variability derived in this study.
       However, the level of bias variability of tropospheric column HCHO data from OMI and TROPOMI derived in this
       study agrees with other recent studies (e.g., Vigouroux et al., 2020; De Smedt et al., 2021) which used other sources
       of evaluation data; therefore, we feel the conclusions drawn here are robust.

**3.3 High pollution case study**

       During the LISTOS 2018 campaign there were large tropospheric column $NO_2$ values retrieved on August 24, 2018
       by both NASA OMI and TROPOMI. This day was also identified as a day of high $NO_2$ pollution concentrations, albeit
       not an $O_3$ exceedance day, during the campaign by Judd et al. (2020). Figure 5 illustrates the values of tropospheric
       FNRs retrieved by NASA OMI and TROPOMI and measured by airborne observations on this day. Figure S4 displays

the spatial distribution of tropospheric column $NO_2$ and HCHO from NASA OMI and TROPOMI and the scatter plot
       comparison to airborne observations. Figure 5 demonstrates that both satellite retrievals and airborne observation data
       observed large areas of VOC/radical-limited $O_3$ regimes (FNR < 1.0) in the NYC region. On this day, tropospheric
       column $NO_2$ values measured by GCAS reached values > $2.0 \times 10^{16}$ molecules cm$^{-2}$ in large portions of the
       VOC/radical-limited regions. Furthermore, when comparing airborne tropospheric column HCHO values on August

24, 2018 (Fig. S4) to campaign-averaged values (Fig. S3), it is clear that HCHO concentrations were lower on this
       day compared to other days throughout the summer. In combination with the large $NO_2$ concentration, this further
       increased the VOC/radical-limitation on this day. The low HCHO/VOC concentrations measured by airborne and
       space-based remote-sensing products throughout the extensive VOC-limited regime could be the reason why a large-
       scale $O_3$ exceedance event was not experienced on August 24, 2018 in proximity to NYC.

**Table 3. Statistical evaluation of NASA OMI and TROPOMI retrievals of tropospheric column $NO_2$ and
       HCHO, and resulting FNRs, on August 24, 2018. Statistics presented are number of co-located grids (N),
       median bias ± bias standard deviation, normalized median bias (NMB, %), coefficient of determination ($R^2$),
       and linear regression slope (Slope).**

| | NASA OMI (0.15° × 0.15°) | | | | TROPOMI (0.05° × 0.05°) | | |
|---|---|---|---|---|---|---|---|
| | FNR | HCHO* | NO₂* | | FNR | HCHO* | NO₂* |
| N | 20 | 20 | 21 | N | 147 | 147 | 154 |
| Bias | 0.1±1.3 | 4.8±4.8 | 3.5±7.2 | Bias | 0.6±1.5 | 4.7±6.3 | -0.6±8.8 |
| NMB | 9.6 | 66.1 | 28.5 | NMB | 40.9 | 56.3 | -4.3 |
| $R^2$ | 0.35 | 0.25 | 0.65 | $R^2$ | 0.32 | 0.03 | 0.73 |
| Slope | 0.89 | 1.17 | 0.49 | Slope | 0.80 | 0.33 | 0.42 |

*bias units are $\times 10^{15}$ molecules cm$^{-2}$.

NASA OMI retrievals in the region of lowest FNRs (40.5°N – 41.0°N, 74.0°W – 73.5°W) compared well to
       observations. In this region, average NASA OMI tropospheric FNRs (0.84) and GCAS observations at 0.15° × 0.15°
       spatial resolution (1.00) were both ≤ 1.0. TROPOMI retrievals resulted in slightly larger average FNRs (1.15) in this





area of VOC/radical-limited regions compared to GCAS observations at $0.05° \times 0.05°$ spatial resolution (0.76). Both NASA OMI and TROPOMI had similar median biases ($\sim 5.0 \times 10^{15}$ molecules cm$^{-2}$) when compared to observed

tropospheric HCHO; however, TROPOMI had a smaller median bias ($-0.6 \pm 8.8 \times 10^{15}$ molecules cm$^{-2}$) compared to NASA OMI ($3.5 \pm 7.2 \times 10^{15}$ molecules cm$^{-2}$) when evaluated with measured tropospheric NO$_2$ data (see Table 3). This lower median bias in TROPOMI retrieved tropospheric column NO$_2$ compared to the same sensor's HCHO statistics led to the slight high bias compared to observed FNRs ($0.6 \pm 1.5$). The similar high biases in NASA OMI tropospheric column NO$_2$ and HCHO resulted in FNRs which compared well to the observed values (median bias = $0.1 \pm 1.3$). This

further emphasizes the important result of this study that when investigating satellite retrievals of FNRs, that accurate FNRs do not necessarily mean that the particular satellite sensor accurately retrieves both NO$_2$ and HCHO. For instance, NASA OMI has a NMB of < 10% when compared to GCAS FNRs; however, both tropospheric column NO$_2$ and HCHO have NMB values of 28.5% and 66.1%, respectively. Furthermore, both NASA OMI and TROPOMI have FNR linear regression slopes $\geq 0.8$, suggesting accurate retrievals; however, tropospheric column NO$_2$ and HCHO

data from both satellite products have linear regression slopes which largely deviate from unity (see Fig. S4). Therefore, one should have caution when assuming satellite-retrieved accuracy of FNRs as offsetting biases in NO$_2$ and HCHO can mask errors in both, or individual, retrieved products.

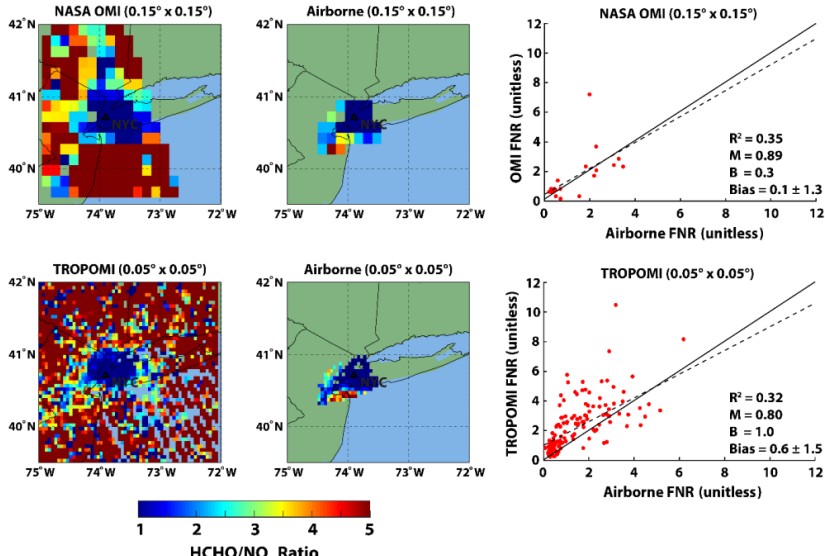

**Figure 5: NASA OMI, TROPOMI, and airborne tropospheric column FNR retrievals on August 24, 2018. All co-located**
**satellite and airborne remote-sensing tropospheric column FNR values are averaged at $0.15° \times 0.15°$ for the OMI inter-comparison and $0.05° \times 0.05°$ spatial resolution for TROPOMI. The black triangle indicates the location of the city of NYC. The direct comparison of co-located NASA OMI and TROPOMI FNR data to airborne observations is shown in the scatter plots (right column). The solid black line shows the 1:1 comparison and the dashed line shows the linear regression fit of the comparison. The figure inset shows the main statistics (coefficient of determination ($R^2$), slope (M), y-intercept (B), and**
**median bias and bias standard deviation) of the comparison of satellite and airborne tropospheric column FNR data.**

A major challenge for accurately retrieving tropospheric FNRs with satellite sensors to evaluated O$_3$ sensitivity production regimes is the noise in daily retrievals of HCHO due to low signal-to-noise ratios and low measurement sensitivity of shorter UV/VIS wavelengths to HCHO in the troposphere. The noise in both NASA OMI





and TROPOMI tropospheric column HCHO data on August 24, 2018 can be seen in Fig. S4. Both NASA OMI and
TROPOMI HCHO retrievals display low correlation values when compared to observations (see Table 3). Despite
TROPOMI having higher correlation with observed $NO_2$ compared to NASA OMI, the very low correlation of
TROPOMI with observed HCHO results in lower correlation and higher bias standard deviations of FNRs compared
to NASA OMI. This further emphasizes that the large bias variability, due to noisy data, in retrievals of tropospheric
column HCHO are the limiting factor in using spaceborne observations of daily FNRs.

**3.4 Common a priori sensitivity test**

This section analyzes the impact of using common, high spatial resolution ($4 \times 4$ km$^2$), WRF-CMAQ-predicted $NO_2$
and HCHO vertical profiles as a prior information in NASA OMI and TROPOMI retrievals. GeoTASO and GCAS
retrievals were not reprocessed in order to have a consistent reference data set for satellite evaluation. Figure 6 shows
the campaign-averaged FNRs from NASA OMI and TROPOMI retrievals, when reprocessed with WRF-CMAQ $NO_2$
and HCHO a priori vertical profiles, compared to co-located airborne remote-sensing products (scatter plot
comparison displayed in Fig. S5; statistical evaluation shown in Table 4). Comparing NASA OMI FNRs from this
figure to Fig. 1, it is evident that using high spatial resolution WRF-CMAQ-predicted $NO_2$ and HCHO vertical profiles
as a prior information resulted in FNR retrievals that are better able to capture the low FNR values (FNR $\leq$ 1.0)
observed around NYC. Reprocessed TROPOMI FNRs also have lower values around NYC; however, were reduced
less compared to OMI retrievals. Furthermore, when comparing the results in Fig. S5 to Fig. 4 further demonstrates
how the reprocessed satellite retrievals better capture the lower FNR values (FNR < 2.0).

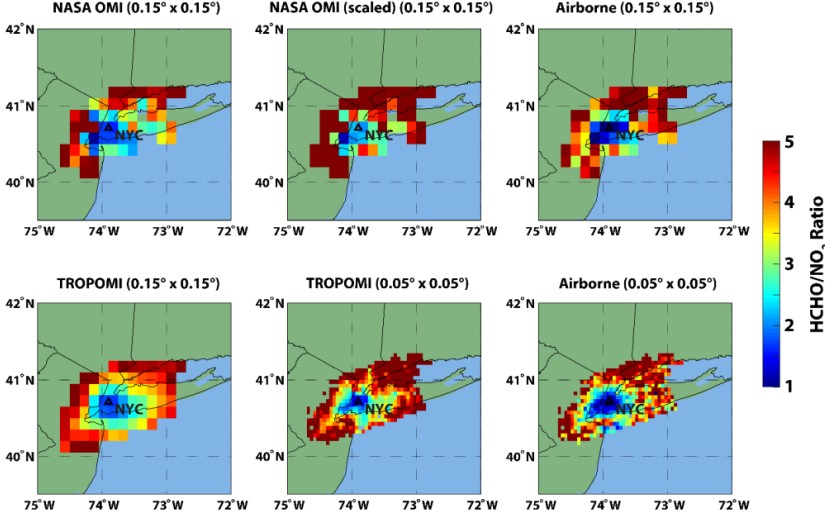

**Figure 6: NASA OMI and TROPOMI reprocessed tropospheric column FNR retrievals compared to airborne FNR
observations averaged for all flights. All co-located OMI and airborne remote-sensing tropospheric column FNR values are
averaged at $0.15° \times 0.15°$ and TROPOMI co-locations are averaged at both $0.15° \times 0.15°$ and $0.05° \times 0.05°$ spatial resolution.
The OMI FNR retrievals calculated with the scaled WRF-CMAQ profiles are identified as "scaled". The black triangle
indicates the location of the city of NYC.**





Comparing standard retrieval products from NASA OMI (see Fig. S2 for $NO_2$ and Fig. S3 for HCHO) to reprocessed retrievals using WRF-CMAQ a priori profiles (see Fig. S6 for $NO_2$ and Fig. S7 for HCHO), it is clear that

in general the higher spatial resolution model data resulted in larger tropospheric column $NO_2$ and slightly larger tropospheric column HCHO values. For TROPOMI, reprocessing the retrievals with WRF-CMAQ a priori information caused increases in tropospheric column $NO_2$ over polluted regions, but small decreases over rural areas characterized by background concentrations. Tropospheric column HCHO data for the reprocessed TROPOMI data were slightly lower in more polluted urban regions near NYC and much lower in the rural areas dominated by

background concentrations compared to standard retrievals.

The increases in NASA OMI tropospheric column $NO_2$ concentrations resulted in a small negative bias in FNR retrievals (-0.3±3.9), compared to a small positive bias in the standard products (0.4±3.8). When compared to airborne observations the reprocessed NASA OMI tropospheric column $NO_2$ data displays a large positive median bias ($3.1 \pm 5.1 \times 10^{15}$ molecules cm$^{-2}$) which was not evident in the standard retrieval products. Similarly, for evaluation

of the reprocessed NASA OMI tropospheric column HCHO data, a higher positive bias ($8.6 \pm 7.8 \times 10^{15}$ molecules cm$^{-2}$) was calculated compared to observations. It should be noted, as previously discussed, that offsetting high biases in both reprocessed NASA OMI tropospheric column $NO_2$ and HCHO retrievals resulted in mean FNR values that compared relatively well to observations.

**Table 4. Statistical evaluation of NASA OMI and TROPOMI retrievals of tropospheric column $NO_2$ and HCHO, and resulting FNRs, when reprocessed with high spatial resolution WRF-CMAQ a prior information. Statistics presented are number of co-located grids (N), median bias ± bias standard deviation, normalized median bias (NMB, %), coefficient of determination ($R^2$), and linear regression slope (Slope).**

| NASA OMI (0.15° × 0.15°) | | | | Scaled NASA OMI (0.15° × 0.15°)[1] | | | |
|---|---|---|---|---|---|---|---|
| | FNR | HCHO* | NO₂* | | FNR | HCHO* | NO₂* |
| N | 101 | 101 | 116 | N | 101 | 101 | 116 |
| Bias | -0.3±3.9 | 8.6±7.8 | 3.1±5.1 | Bias | 0.5±3.2 | 4.4±7.1 | -0.3±3.9 |
| NMB | -9.4 | 65.7 | 50.0 | NMB | 16.7 | 35.6 | -4.2 |
| R² | 0.17 | 0.30 | 0.65 | R² | 0.21 | 0.25 | 0.67 |
| Slope | 0.85 | 0.70 | 1.03 | Slope | 1.05 | 0.50 | 0.76 |
| TROPOMI (0.15° × 0.15°) | | | | TROPOMI (0.05° × 0.05°) | | | |
| | FNR | HCHO* | NO₂* | | FNR | HCHO* | NO₂* |
| N | 261 | 261 | 261 | N | 1693 | 1741 | 1802 |
| Bias | -0.3±1.4 | -1.2±5.1 | 0.1±3.8 | Bias | 0.2±2.2 | -0.1±6.3 | -0.4±4.1 |
| NMB | -9.1 | -9.4 | 2.0 | NMB | 4.7 | -0.3 | -6.4 |
| R² | 0.43 | 0.35 | 0.61 | R² | 0.32 | 0.32 | 0.67 |
| Slope | 0.67 | 0.41 | 0.55 | Slope | 0.74 | 0.58 | 0.61 |

*bias units are ×10$^{15}$ molecules cm$^{-2}$.

[1]reprocessed with "scaled" CMAQ a priori profiles.

The larger tropospheric column $NO_2$ concentrations in reprocessed NASA OMI data using high spatial resolution model data as a priori information was also shown in past studies (e.g., Souri et al., 2016; Goldberg et al.,



2017). Both our study and the work by Goldberg et al. (2017) show that high spatial resolution CMAQ-predicted $NO_2$ a priori profiles results in OMI tropospheric column $NO_2$ concentrations that are as high as a factor of 2 larger than the standard retrievals. This high bias is caused by smaller AMFs calculated due to the shape factor of high spatial

resolution CMAQ-predicted $NO_2$ concentrations having a too steep $NO_2$ gradient. The steeper shape factor is caused by higher $NO_2$ concentrations in the PBL and lower values in the free troposphere compared to the a priori profiles used in standard NASA OMI retrievals. The change in HCHO shape factors when using WRF-CMAQ a priori profiles resulted in slightly higher tropospheric column HCHO concentrations when compared to standard products for the same reason as tropospheric column $NO_2$. Similar to Goldberg et al. (2017), we used airborne in situ observations of

$NO_2$ and HCHO from LISTOS 2018 and the Ozone Water-Land Environmental Transition Study 2 (OWLETS-2, https://www-air.larc.nasa.gov/missions/owlets/) field campaigns, OWLETS-2 took place just prior to LISTOS-2018 during the summer of 2018 in the Baltimore, MD region, to correct the model-predicted a priori profiles for use in NASA OMI retrievals and is discussed later in this section.

TROPOMI reprocessed retrievals at $0.05° \times 0.05°$ spatial resolution displayed improved performance when
compared to all standard retrieval products of HCHO and FNR. Tropospheric column $NO_2$ concentrations in reprocessed TROPOMI retrievals resulted in a slightly lower median biases ($-0.4\pm4.1 \times 10^{15}$ molecules $cm^{-2}$) compared to the standard products ($-0.3\pm3.7 \times 10^{15}$ molecules $cm^{-2}$). Reprocessing TROPOMI retrievals of tropospheric column HCHO resulted in smaller concentrations and much improved median biases and bias standard deviation ($-0.1\pm6.3 \times 10^{15}$ molecules $cm^{-2}$) compared to the standard products ($1.9\pm6.7 \times 10^{15}$ molecules $cm^{-2}$). The
good performance of both reprocessed TROPOMI $NO_2$ and HCHO resulted in FNR values with a smaller median bias when evaluated with observations ($0.2\pm2.2$) compared to standard products ($0.4\pm2.3$).

As mentioned earlier, when WRF-CMAQ-predicted a priori profiles were used in NASA OMI retrievals it resulted in smaller AMF calculations compared to standard products, resulting in larger tropospheric column $NO_2$ and HCHO concentrations and higher biases when evaluated with observations. Following methods similar to Goldberg
et al. (2017) we used the University of Maryland (UMD) Cessna 402B airborne observations to apply in situ data observational constraints on the $NO_2$ and HCHO a priori profiles applied in NASA OMI retrievals. The evaluation of WRF-CMAQ-predicted $NO_2$ (14 flights during LISTOS 2018 and OWLETS-2) and HCHO (7 flights during LISTOS 2018) vertical profiles using airborne data is displayed in Fig. S8. The comparison of WRF-CMAQ-predicted $NO_2$ concentrations to airborne in situ observations emphasizes how the a priori profile vertical gradients from the model
runs are too steep. Compared to measured $NO_2$ values, the model displays a high bias below 1 km agl of ~0.4 ppb which was often > 50% larger than observations. This is in stark contrast to the model performance above 2 km agl where the model has a low bias of -0.2 to -0.4 ppb often approaching 100% lower than observations. For the WRF-CMAQ comparison to airborne in situ HCHO data, the model has a low bias throughout the lower troposphere, with larger low biases near the surface (-3.0 ppb between 0-1 km agl) and smaller low biases in the free troposphere (~-1.3
ppb above 2 km agl). These low biases range between -50 to -100% lower compared to measured values. In addition to biases in emission inventories, chemical mechanisms, and other physiochemical parameterizations applied in CTMs, meteorological predictions by WRF, such as wind speed and direction, must have limited errors in order to accurately predict the horizontal and vertical distribution of $NO_2$ and HCHO concentration (e.g., Laughner et al., 2016;



Liu et al., 2021). Compared to the airborne in situ observations taken during LISTOS 2018 and OWLETS-2, WRF wind speed and direction predictions during this study performed relatively well with median correlation (R) and bias values of 0.70 and 0.63 and ≤1.0 m s$^{-1}$ in the u- and v-wind components, respectively. While the WRF simulations applied in this study capture the spatiotemporal variability and general magnitude of observed wind speed and direction, this does not mean that simulated meteorology did not partially contribute to the errors in vertical NO$_2$ and HCHO profiles simulated by WRF-CMAQ.

Using this model evaluation, we applied approximated scaling factors to the a priori profiles to reprocess NASA OMI data (hereinafter referred to as "scaled"). Separate scaling factors were applied above and below the PBL, approximated to be at 1.5 km agl, where noticeable differences in model performance were evident. For NO$_2$, the model displays a high bias in the PBL and a low bias in the free troposphere and we apply a scaling factor of 0.5 to WRF-CMAQ a priori NO$_2$ profiles in the PBL and 5.0 above the PBL. For HCHO, WRF-CMAQ predictions displayed

low biases throughout the lower troposphere, and we applied a scaling factor of 2.0 to WRF-CMAQ a priori profiles in the PBL and 5.0 above the PBL. These scaling factors are approximations of the model performance and are simply applied to determine the impact of "raw" and "scaled" WRF-CMAQ-simulated a priori profiles in NASA OMI NO$_2$ and HCHO retrievals. Since the UMD Cessna 402B in situ data have limited spatiotemporal coverage of the LISTOS-2018 and OWLETS-2 domains, we did not want to apply overly specific scaling factors to represent all locations/times

studied in this work.

The spatial distribution of FNRs derived from the scaled NASA OMI reprocessed NO$_2$ and HCHO retrievals is shown in Fig. 6 (scatter plot comparison displayed in Fig. S5; statistical evaluation shown in Table 4). From Table 4 and Fig. 6 it can be seen that the scaled WRF-CMAQ a priori profiles result in higher FNR values and improved tropospheric column NASA OMI NO$_2$ and HCHO retrievals compared to reprocessed products using the raw model

output (see Fig. S6 and S7). Scaled NASA OMI tropospheric column NO$_2$ and HCHO retrievals had much smaller biases and bias standard deviations of -0.3±3.9 × 10$^{15}$ molecules cm$^{-2}$ and 4.4±7.1 × 10$^{15}$ molecules cm$^{-2}$, respectively, compared to the retrievals with raw WRF-CMAQ predictions. This result demonstrates the need for accurate shape factors (i.e., vertical distribution of trace gases) to be used as a priori information in NASA OMI retrievals. Finally, the improved accuracy of tropospheric column NO$_2$ and HCHO retrievals resulted in a slightly higher FNR median

bias (0.5±3.2) compared to reprocessed data using raw CMAQ predictions.

### 3.5 Expected FNR information from TEMPO

TEMPO is expected to provide revolutionary information about air quality in North America (Zoogman et al., 2017; Chance et al., 2019). This geostationary sensor will provide tropospheric column NO$_2$ and HCHO data, and resulting FNR products, up to every 1-2 hours during daylight hours. Here we demonstrate the expected improvement in the

information content of tropospheric FNRs due to the diurnal retrievals of TEMPO compared to low earth orbit sensors (e.g., OMI and TROPOMI) which retrieve a single snapshot at ~13:30 local time. Synthetic OMI and TROPOMI FNRs are derived here by averaging the synthetic TEMPO data at 0.13° × 0.25° and 0.07° × 0.05°, respectively (representative of these sensor's native spatial resolution). This was done to provide synthetic FNR data from the three sensors which only differ based on the spatiotemporal sampling frequency and not retrieval specifics. Once TEMPO





is launched, studies should be conducted to determine the difference in tropospheric FNR values due to the actual
       retrieval products between this geostationary sensor and other low earth orbit satellites.

            One of the main improvements in tropospheric FNR information that is expected from TEMPO compared to
       low earth orbit sensors is the additional data throughout the diurnal cycle. Figure 7a shows the spatial distribution of
       2-hour-averaged synthetic TEMPO FNRs averaged at $0.03° \times 0.05°$ spatial resolution (representative of the native
spatial resolution of TEMPO) on July 12, 2020. This day was chosen due to data availability and the limited cloud
       coverage simulated for the synthetic product on this day. This figure also shows the time series of FNRs retrieved by
       TEMPO, and 13:30 retrieved OMI and TROPOMI FNRs, averaged for an urban region within 0.25° of NYC (Fig. 7b)
       and a rural region 1° north and west of NYC. It is clear that significant information about tropospheric column $NO_2$
       and HCHO, and resulting FNRs, which will be used to investigate $O_3$ production sensitivity regimes will be gained
when TEMPO is launched compared to OMI and TROPOMI. Due to emissions and chemical production/destruction
       of $NO_2$ and HCHO, tropospheric FNRs vary significantly throughout the day (by around a factor of 3 in the vicinity
       of NYC on this day, see Fig. 7b) with large swaths of VOC/radical-limited regions in the northeast US in the morning
       hours which transition to $NO_x$-limited regimes during the afternoon. Rural regions also display a strong diurnal pattern
       of FNRs; however, with much higher values compared to urban regions due to the lack of significant $NO_x$ emission
sources (see Fig. 7c). During the afternoon hours when overpasses of OMI and TROPOMI occur, FNRs are larger
       compared to morning and evening values which will be retrieved by TEMPO. In addition to the improved temporal
       resolution of TEMPO, the increased spatial resolution of these retrievals compared to OMI and TROPOMI will also
       provide improved information on spatial distributions of these proxy species and FNRs.



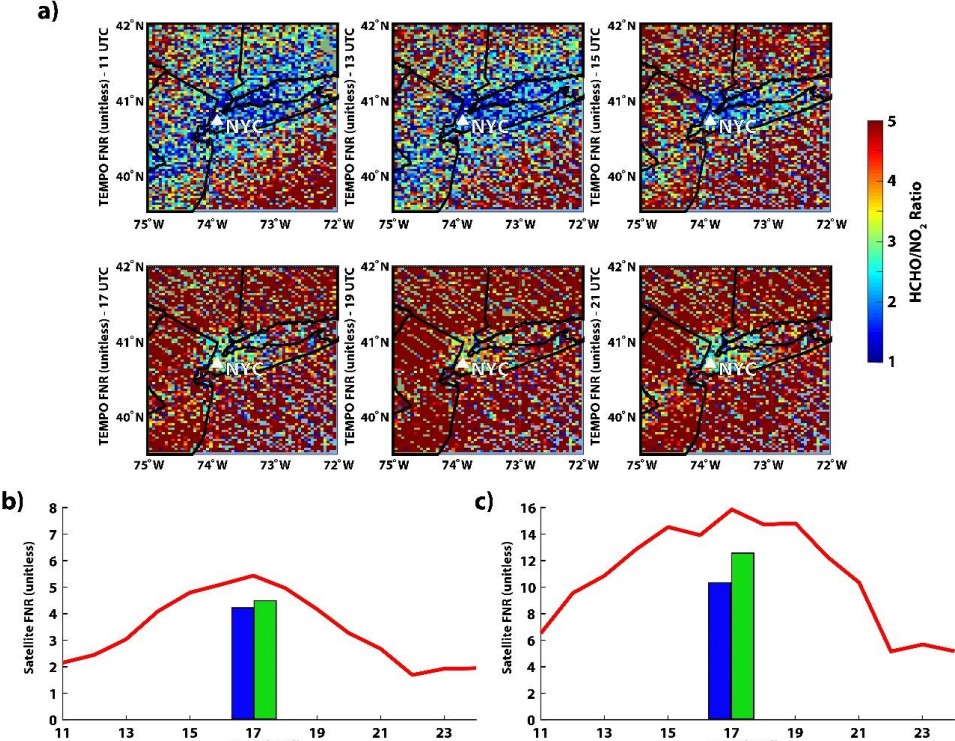

**Figure 7: a) Synthetic TEMPO 2-hour-averaged tropospheric column FNR retrievals between 11 and 21 UTC on July 12, 2020. All synthetic TEMPO tropospheric column FNR data are averaged to a 0.03° × 0.05° spatial resolution. The white triangle indicates the location of the city of NYC. This figure also shows the timeseries of hourly TEMPO FNR values (red line between 11 on July 12, 2020 and 00 UTC on July 13, 2020) averaged b) within 0.25° of NYC and c) a rural location 1° north and west of NYC. The value of tropospheric column FNR retrieved by synthetic OMI (blue bar) and TROPOMI (green bar) at ~13:30 local time (represented by 17 UTC data) are presented in panel b) and c).**

## 4 Conclusions

This study presents a statistical evaluation and inter-comparison of tropospheric FNR retrievals from two commonly applied low earth orbit sensors for investigating $O_3$ production sensitivity regimes (i.e., OMI and TROPOMI). The evaluation of NASA OMI, QA4ECV OMI, and TROPOMI retrievals of tropospheric $NO_2$ and HCHO, and resulting FNRs, was conducted with airborne remote-sensing observations (GeoTASO and GCAS) during LISTOS 2018. Past studies have focused on the evaluation of satellite retrievals of tropospheric column $NO_2$ and HCHO, individually; however, this is the first study to validate and inter-compare multiple satellite platform's and retrieval algorithm's ability to retrieve tropospheric FNRs and also quantify the impact of horizontal spatial resolution, a priori vertical profile information, and different retrieval algorithms. The quantification of satellite-retrieved tropospheric FNRs biases/errors is currently an important, but relatively unknown, uncertainty when applying spaceborne remote-sensing products to investigate $O_3$ production regimes.





NASA OMI, QA4ECV OMI, and TROPOMI retrievals reproduce this general spatial pattern of observed tropospheric FNRs; however, displayed higher FNRs (between 1.0 and 3.0) in the urban regions of NYC where observations suggest $NO_x$-saturated regimes (FNR < 1.0). The statistical evaluation of these satellite products illustrated that all three retrievals have a high bias of background-level tropospheric column $NO_2$ and HCHO concentrations. The satellite retrievals compare more accurately to larger tropospheric column $NO_2$ and HCHO values observed in the moderately polluted areas with a tendency towards a low bias in the more polluted areas. The magnitude-dependent biases for OMI and TROPOMI $NO_2$ and HCHO derived in this study agrees with other recent validation projects (e.g., Judd et al., 2020; Vigouroux et al., 2020; Compernolle et al., 2020; Lamsal et al., 2021; De Smedt et al., 2021). Both OMI and TROPOMI retrievals compared well to observed $NO_2$ throughout the campaign with NMB values < 10%. The statistical comparison with observed HCHO data resulted in larger and more variable biases between the three satellite products. Overall, daily and campaign-averaged comparisons of the satellite HCHO data to observations displayed large bias standard deviations emphasizing the large noise in these retrieval products which hinders the accuracy of FNRs from spaceborne sensors. Averaging TROPOMI HCHO data to coarser spatial resolutions, in order to improve signal-to-noise, proved capable to reduce bias standard deviations compared to observations. While all three satellite products at the near native spatial resolutions had campaign-averaged FNR median biases < 0.5, the bias standard deviations were high (> 2.0), primarily due to noise in the HCHO retrievals. Given the limited measurement sensitivity of shorter UV/VIS wavelengths to HCHO in the middle to lower troposphere, improved information (in situ, remote-sensing, or models) of the vertical profiles of HCHO to be used as a priori information would benefit satellite remote-sensing capabilities for observing HCHO and FNRs.

The higher spatial resolution of TROPOMI, along with a good signal-to-noise ratio, allows this sensor to better capture the spatiotemporal variability and urban/rural interface of tropospheric column $NO_2$ and HCHO values and resulting FNRs. This satellite data had the highest correlations with observed $NO_2$, HCHO, and FNRs throughout the campaign, along with lowest bias standard deviation of all three satellite products. The added benefit of TROPOMI spatial resolution is important as this sensor has now been operational for 5 years and can be applied in trend analysis along with case studies. Future studies of FNR trends should include both OMI and TROPOMI retrievals and determine best practices to fuse/link the two data sets.

NASA OMI retrievals of tropospheric FNRs had lower median biases and bias standard deviations compared to TROPOMI on a day identified as having high $NO_2$ pollution (August 24, 2018). However, this sensor did not provide more accurate retrievals of HCHO and $NO_2$ compared to TROPOMI. This fact, along with results from the campaign-averaged analysis, demonstrates that biases in tropospheric $NO_2$ and HCHO can offset resulting in accurate FNR retrievals. While accurate FNR retrievals are informative, and necessary to studying regimes of $O_3$ production sensitivity, the actual magnitudes of tropospheric $NO_2$ and HCHO concentrations are vital for calculating/investigating quantitative $O_3$ production rates (Souri et al., 2022a). Therefore, it is important to understand the accuracy of not only a satellite's FNR retrievals, but also the ability to retrieve the magnitudes of both chemical proxy species. Another interesting finding during this high $NO_2$ event was that all satellite and airborne observations measured a large region where $O_3$ production was likely VOC-limited (FNRs < 1.0) in the vicinity of NYC; however, no large-scale $O_3$ exceedance events occurred on this day. Interestingly, all satellite and airborne observations on this day also measured





lower than average HCHO concentrations in the vicinity of NYC and these low concentrations of HCHO, a proxy for

VOC abundances, in the VOC-limited $O_3$ production regime around NYC could have been the reason why $O_3$ formation was not elevated on this day.

Applying multiple retrieval algorithms to the radiances of a single satellite sensor is of interest in order to determine how input variables (e.g., information on a priori vertical profiles, clouds, surface albedo, etc.) impact the retrieval performance to identify the most accurate data products. In this study we evaluated results of OMI retrievals

applying two well-known retrieval algorithms (i.e., NASA version 4 product and output from the QA4ECV project). Results from the two retrievals were similar for $NO_2$ but differed primarily in tropospheric column HCHO, where NASA OMI data had a median bias a factor of two larger than QA4ECV data. Both retrieval algorithms resulted in high bias standard deviation of tropospheric HCHO. While NASA OMI data displayed less accurate retrievals in HCHO, and similar performance for $NO_2$, compared to QA4ECV data, NASA OMI data resulted in FNR values with

similar median bias and slightly higher bias standard deviations. Given that both the NASA and QA4ECV retrievals of tropospheric HCHO resulted in noisy data products from OMI (illustrated by large bias standard deviations), this emphasizes the need for improved signal-to-noise and calibration and improved a priori vertical profile information of HCHO to negate the low measurement sensitivity of HCHO in the middle to lower troposphere for future satellite sensors and/or improved retrieval algorithms of HCHO. Addressing this issue, a new SAO OMI collection 4 HCHO

retrieval product is planned to be released by the end of 2022 (personal communication with the SAO algorithm team). The new retrieval could represent a step forward in the quality of the OMI HCHO product with improvements in OMI radiance calibration and quality control translating to a more stable and less noisy HCHO retrievals from OMI. Future studies should apply this potentially improved OMI HCHO retrieval product to evaluate the improvement in satellite-derived FNRs.

Our study investigated the impact of high spatial resolution WRF-CMAQ-predicted $NO_2$ and HCHO a priori profiles on OMI and TROPOMI retrievals of FNRs. Using the WRF-CMAQ-predicted a priori information resulted in highly accurate retrievals of FNRs with median biases $\leq 0.5$ over the entire campaign. However, while reprocessed NASA OMI data had only a small low median bias in FNR, the high spatial resolution model data resulted in large high biases in both tropospheric $NO_2$ and HCHO. These high biases are caused by errors in the shape factor imposed

by the model data. We scaled WRF-CMAQ-predicted vertical profiles of a priori $NO_2$ and HCHO using airborne in situ observations which resulted in smaller biases in the traces gas retrievals. This demonstrates the need for accurate shape factors (i.e., vertical distribution of trace gases) to be used as a priori information in OMI retrievals. Furthermore, while high spatial resolution CTM simulations likely better capture spatial heterogeneity of trace gases such as $NO_2$ and HCHO, shape factors imposed by this specific WRF-CMAQ analysis degraded OMI retrieval performance

compared to standard data products which use the coarser resolution GMI model as a the a priori. TROPOMI reprocessed data on the other hand had improved performance when using the higher spatial resolution WRF-CMAQ data as a priori product compared to standard retrievals which apply coarser resolution TM5 output. The fact that TROPOMI native spatial resolution is similar to the WRF-CMAQ resolution used in this study, could have resulted in the better results when reprocessing TROPOMI data compared to OMI. Future studies should investigate the impact

of various spatial resolution a priori profile data sets, ranging from the ~1° × 1° GMI and TM5 model data used for



OMI and TROPOMI, respectively, to much higher resolution air quality model simulations, on the results of reprocessed satellite NO$_2$ and HCHO data.

Overall, the biases, bias standard deviations, and correlations presented in this study can be used in future studies when interpreting the accuracy of OMI and TROPOMI retrievals of FNRs used for investigating O$_3$ sensitivity
regimes applying satellite products. The individual satellite products display varying degrees of capability to retrieve tropospheric FNRs and it is necessary to further validate OMI and TROPOMI retrievals using other field campaign data in different regions of the world to determine regional biases, and identify the primary controlling factors of systematic and random errors (e.g., cloud fraction, surface albedo, spatial resolution, signal-to-noise ratios, a priori information, etc.). A main take away from this study is that it is necessary to statistically evaluate both the tropospheric
FNRs, and the NO$_2$ and HCHO products, individually, as large biases in both NO$_2$ and HCHO satellite products can offset resulting in accurate FNR values. Our study goes beyond investigating median biases, as the noise in satellite retrievals of HCHO result in large bias standard deviations when compared to observations. The large bias variability/noise in tropospheric column HCHO retrievals appear to be the controlling and limiting factor of daily FNR accuracy.

**Acknowledgements**

Matthew Johnson, Sajeev Philip (grant number: 80NSSC20K1182), Rajesh Kumar (grant number: 80NSSC20K1234), Amir Souri (grant number: 80NSSC21K1333), and Jeffrey Geddes (grant number: 80NSSC20K1033) were funded for this work through NASA's Aura Science Team (NNH19ZDA001N-AURAST). Laura Judd and Scott Janz are collaborators on the NASA Aura Science Team project which funded the majority of this work and their contribution
to this study was through in-kind efforts. Sajeev Philip acknowledge support from the NASA Academic Mission Services by Universities Space Research Association at NASA Ames Research Center during the initial stages of this study. Finally, Aaron Naeger is funded through the NASA TEMPO project and his efforts for this study was through in-kind efforts. The authors perceive no financial, or other affiliations, which are conflicts of interest. Resources supporting this work were provided by the NASA High-End Computing (HEC) Program through the NASA Advanced
Supercomputing (NAS) Division at NASA Ames Research Center. The National Center for Atmospheric Research is sponsored by the National Science Foundation. Finally, the views, opinions, and findings contained in this report are those of the authors and should not be construed as an official NASA or United States Government position, policy, or decision.

**Data Availability**

The primary data sources used in this study were the NASA OMI NO$_2$ and SAO HCHO (https://earthdata.nasa.gov/; last access: 4/27/2020), QA4ECV OMI NO$_2$ and HCHO (http://www.qa4ecv.eu/ecvs; last access: 3/3/2020), and TROPOMI PAL NO$_2$ (https://data-portal.s5p-pal.com/; last access: 12/20/2020) and operational HCHO (https://earthdata.nasa.gov/; last access: 4/27/2020) satellite data. For evaluating these satellite products we use airborne remote sensing data from GeoTASO and GCAS which were downloaded from the LISTOS-2018 campaign



data repository (https://www-air.larc.nasa.gov/missions/listos/index.html; last access: 4/21/2020). Finally, the synthetic TEMPO data product was downloaded from: https://weather.msfc.nasa.gov/tempo/data.html; last access: 4/15/2021.



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
