# Peer review of "Satellite remote-sensing capability to assess tropospheric column ratios of formaldehyde and nitrogen dioxide: case study during the LISTOS 2018 field campaign"

_Atmospheric Measurement Techniques, 2022_

## Referee Comment (RC2)

**Review of "Satellite remote-sensing capability to assess tropospheric column ratios of formaldehyde and nitrogen dioxide: case study during the LISTOS 2018 field campaign", Johnson et al., AMTD, 2022.**

Johnson et al. present a detailed statistical analysis of FNR observations with two different OMI products, and with TROPOMI. FNR, NO2 and HCHO satellite retrievals are validated against airborne measurements over the New-York area during summer 2018. It is well demonstrated in the paper that the noise of the HCHO satellite retrievals is the limiting factor for the FNR observations since the individual HCHO columns need to be averaged at poorer time and space resolution than NO2. The precision of the OMI HCHO observations does not allow for daily FNR observations at OMI native resolution. The OMI QA4ECV HCHO product is found to perform better than the OMI NASA HCHO product. TROPOMI offers an important improvement in the spatial and temporal resolution of HCHO and NO2 tropospheric columns, allowing for daily FNRs retrievals at TROPOMI native spatial resolution. The results are further improved by averaging TROPOMI observations on a larger spatial grid. Both NO2 and HCHO satellite products suffer from bias compared to aircraft observations. This study identifies an important positive bias over rural regions (lowest columns) for both species, and for OMI and TROPOMI products. However, the positive bias found for the TROPOMI products is reduced compared to OMI thanks to the better spatial resolution and lower noise. It is also demonstrated that the bias of the FNR satellite observations is much lower than the respective NO2 and HCHO biases. This is an important result that would deserve more discussion in the paper. The paper is well written, albeit a bit long and too detailed. The scientific approach is solid, however some points should be tested or clarified. I recommend publication in AMT after some revisions.

**General comments**

One concern is the small number of days that are available for the validation. Here the field campaign covers only a few days (8 days collocated with OMI, 12 days with TROPOMI). The statistical results are not always significant, especially for OMI. Studies on longer time period could improve the observed correlations, that are poor for HCHO.

- The paper could be improved by providing an information about the spatial and temporal resolution that might provide useful FNR observations with OMI (ex. monthly averaged data). How many observations are needed at minimum to reduce the noise at the level of the TROPOMI daily observations?
- It would be interesting to know if the HCHO observations with aircraft instruments are also noisier than the NO2 observations, and therefore also the limiting factor of suborbital FNR observations.
- In Table 2, I recommend adding a line providing the mean value +- the standard deviation of FNR, HCHO and NO2 for the aircraft, NASA OMI, QA4ECV OMI, and TROPOMI (0.15° and 0.05°).
- I recommend more tests on the selection of the data, that is currently at the edge of the statistical significance (see later).

One interesting result of the paper is that the errors in NO2 and HCHO columns tend to offset in the FNR observations. There might be good reasons for this, such as error cancellation. It is therefore important to use HCHO and NO2 products that have been retrieved with algorithms and auxiliary data as consistent as possible. This is an important message for the future TEMPO product.

- It would be good to discuss further what type of error might cancel out, or at least might reduce, when using NO2 and HCHO retrieved using consistent algorithms to derive FNR (surface albedo, cloud products, a priori profiles).
- I recommend adding a table providing a quick look at the auxiliary data used in the AMF calculations for the NASA, QA4ECV and TROPOMI products, and TEMPO.
- Discuss the different FNR biases with the level of consistency between NO2 and HCHO AMF settings.

The low HCHO correlations are also partly due to lower spatial variability of the HCHO distribution compared to NO2, also in the airborne measurements, over the time and domain of the study.

Selection of data:

- Filter row anomaly both for HCHO and NO2 products.
- The lower bound limits for HCHO and NO2 appear to bestrict, compared to the reported standard deviations of the bias. For HCHO, the bias std ranges from 9 to 5e15 molec.cm-2, while the lower limit has been set to -8e15. For NO2, bias std is about 4e15, while the lower limit has been set to -1e15 molec.cm-2. There is a possibility that a significant part of the negative values has been filtered out while it actually belongs to the normal distribution. The effect could be an artificial increase of the mean background values. Please test a lower bound limit for the data selection.
- At the spatio-temporal resolution of the study, OMI retrievals are clearly at their detection limit. Please consider testing a lower grid resolution (0.2°) for OMI.
- To increase the number of collocations, I would suggest testing a larger temporal window of 3h for the airborne retrievals.

It would be good to better stress the specificities of this paper compared to the recent paper of Souri et al., 2022, which also compares OMI and TROPOMI NO2, HCHO and FNR errors over the US. (Characterization of Errors in Satellite-based HCHO / NO2 Tropospheric Column Ratios with Respect to Chemistry, Column to PBL Translation, Spatial Representation, and Retrieval Uncertainties)

**Detailed comments**
**Abstract**
Line25: "high spatiotemporal coverage": please provide numbers, such as the native resolution of OMI and TROPOMI. I would rephrase "OMI and TROPOMI are capable of providing NO2 and HCHO daily global observation at native resolution of respectively … and …". However, satellite observations are known to be affected by noise and biases, that limit the precision of FNR.

Line 25: "…, yet a recent study suggested ….". This sentence is rather vague. Which study?

Line 30: Please specify the covered period.

Line 32: Please be clearer in the abstract with the term "suborbital". This is not obvious for a general reader.

Line 49: Place replace large by larger biases.

**Introduction**
Line 95: please add the 2 following references: Wang et al, 2022; Harkey et al., 2015.

Line 100: the choice of references seems weird. It might be good to add references for NO2 and HCHO L2 products of each sensor, and not only for studies using both species together. The SCIAMACHY instrument is missing in the list.

**Methods**

Line 163: The OMI rows affected by the row anomaly should be filtered out in the HCHO product such as in the NO2 product. The reference sector method does not correct for the row anomaly, but for the stripes between the valid rows. Please rephrase (and check that the HCHO data are filtered correctly).

Line 204: Please explain what you mean by "iterative fitting algorithm" and "simultaneous fitting". To me, a DOAS fit is an iterative fit (least-squared fit).

Line 215: The QA4ECV fitting window is 328.5-359 nm, such as TROPOMI. For all HCHO products, please double check the retrieval intervals that are mentioned in the paper. Most of the recent retrievals use a fitting window larger than 328.5-346 nm.

Line 261: Please explicit the term SWs.

**Results**

Line 444: "Tropospheric columns NO2 concentrations", "tropospheric columns NO2 retrievals". Could be simplified to "Tropospheric NO2 columns" and homogenized throughout the paper.

Line 465: It should be emphasized here that TROPOMI offset for low columns is lower than OMI at the resolution of 0.05.

Line 491: add a reference to Verhoelst et al. 2021.

To our knowledge, the cited references do not report a high bias of NO2 for background values. But the studies were made with the previous version of the TROPOMI NO2 product. This should be clarified here.

Line 495. The comparison of TROPOMI NO2 Bias at 0.05 and 0.15° also clearly shows the spatial resolution effect on the background values (from negative to positive and similar to OMI NMB). Please mention this resolution effect.

Table2: Please add one line with the mean FNR, NO2 and HCHO columns and their standard deviations.

Figure 3: Please test different data selection as suggested in the general comments.

Line 517: The results are not so much in agreement with the study of Vigouroux, who reported indeed a high bias for the lowest columns, but for columns lower than 2.5e15 molec.cm-2. The TROPOMI bias ranges from 0 to negative values for columns larger than 5e15 molec.cm-2.

Line 527: Please also compare the bias standard deviation between OMI and TROPOMI.

Line 530: In De Smedt 2021, it is reported that the OMI HCHO offset is larger than for TROPOMI. But the reported bias are all negative for columns larger than 5e15 molec.cm-2. The conclusions of this study are therefore not completely in agreement with De Smedt et al. or with Vigouroux et al..

Line 594: I agree with the reasons for the poor HCHO correlation. Please add that they are also partly due to the low HCHO variability over the studied time and domain. A full year study would result in larger correlations.

High pollution case study: The added value of this section is not clear. As the paper is already long and detailed, I would suggest removing this section. If not removed, I then suggest to discuss the causes of higher NO2 columns and lower HCHO columns, such as surface temperature.

Common a priori sensitivity tests:

- It is not clear why the WRF-CMAQ profiles need to be scaled for the NASA OMI datasets, but not for the TROPOMI datasets.
- Figure 6: please explain in the legend what is the NASA OMI (scaled).
- Comparing Table 2 and Table 4, I can only see an improvement for TROPOMI at 0.05° resolution. The added value of this section is not clear, given the uncertainties in the WRF-CMAQ profiles.

Expected FNR information from TEMPO:

- What is the expected signal ratio of Tempo compared to TROPOMI for NO2 and HCHO? Can we expect an improvement of the HCHO noise?
- It would be interesting to show the diurnal variation of NO2 and HCHO from the TEMPO simulations.
- Line 777 and figure 7b and 7c. Not clear if retrieved OMI and TROPOMI are shown (line 777) or only synthetic TEMPO data averaged at the different spatial resolutions. It should be possible to show real data for OMI and TROPOMI in 2020.

**Conclusion**

Line 831: Please comment on the spatial and temporal resolution allowed by the OMI datasets. This is important for trend studies.

Line 860. The statements made on the new version of the NASA OMI HCHO product appear to be optimistic. The SNR of the retrievals is primarily determined by the SNR of the instrument. Please be more cautious, especially since no publication can support the statements.

Line 866-867: This does not seem so clear in the paper that "using the WRF-CMAQ-predicted a priori information, resulted in highly accurate retrievals of FNRs". All L2 products used in the study also results in median biases lower than 0.5 for FNRs.

Line 871-872: This sentence is misleading. The need for accurate shape factors is not only for OMI retrievals. It should be even more important for TROPOMI and TEMPO because of their finer spatial resolution.

**References**

Verhoelst, T., Compernolle, S., Pinardi, G., Lambert, J.-C., Eskes, H. J., Eichmann, K.-U., Fjæraa, A. M., Granville, J., Niemeijer, S., Cede, A., Tiefengraber, M., Hendrick, F., Pazmiño, A., Bais, A., Bazureau, A., Boersma, K. F., Bognar, K., Dehn, A., Donner, S., Elokhov, A., Gebetsberger, M., Goutail, F., Grutter de la Mora, M., Gruzdev, A., Gratsea, M., Hansen, G. H., Irie, H., Jepsen, N., Kanaya, Y., Karagkiozidis, D., Kivi, R., Kreher, K., Levelt, P. F., Liu, C., Müller, M., Navarro Comas, M., Piters, A. J. M., Pommereau, J.-P., Portafaix, T., Prados-Roman, C., Puentedura, O., Querel, R., Remmers, J., Richter, A., Rimmer, J., Rivera Cárdenas, C., Saavedra de Miguel, L., Sinyakov, V. P., Stremme, W., Strong, K., Van Roozendael, M., Veefkind, J. P., Wagner, T., Wittrock, F., Yela González, M., and Zehner, C.: Ground-based validation of the Copernicus Sentinel-5P TROPOMI NO2 measurements with the NDACC ZSL-DOAS, MAX-DOAS and Pandonia global networks, Atmos. Meas. Tech., 14, 481–510, https://doi.org/10.5194/amt-14-481-2021, 2021.

Wang, P.; Holloway, T.; Bindl, M.; Harkey, M.; De Smedt, I. Ambient Formaldehyde over the United States from Ground-Based (AQS) and Satellite (OMI) Observations. Remote Sens. 2022, 14, 2191. https://doi.org/10.3390/rs14092191

Harkey, M.; Holloway, T.; Oberman, J.; Scotty, E. An evaluation of CMAQ NO2 using observed chemistry-meteorology correlations. J. Geophys. Res. Atmos. 2015, 120, 11775–11797. [CrossRef]

Souri, A. H., Johnson, M. S., Wolfe, G. M., Crawford, J. H., Fried, A., Wisthaler, A., Brune, W. H., Blake, D. R., Weinheimer, A. J., Verhoelst, T., Compernolle, S., Pinardi, G., Vigouroux, C., Langerock, B., Choi, S., Lamsal, L., Zhu, L., Sun, S., Cohen, R. C., Min, K.-E., Cho, C., Philip, S., Liu, X., and Chance, K.: Characterization of Errors in Satellite-based HCHO / NO2 Tropospheric Column Ratios with Respect to Chemistry, Column to PBL Translation, Spatial Representation, and Retrieval Uncertainties, Atmos. Chem. Phys. Discuss. [preprint], https://doi.org/10.5194/acp-2022-410, in review, 2022.

---

## Author Comment (AC1)

**Response to Reviewer #1**

We thank Reviewer #1 for reviewing the manuscript and for their helpful comments. We agree with the majority of the reviewer's feedback and feel that these comments have led to an improvement in the quality of the manuscript. All reviewer comments are in italics and the author's responses are in standard font.

This study evaluates OMI and TROPOMI retrievals of NO2, HCHO and FNR using aircraft measurements during the LISTOS campaign. The manuscript is well-written, and it is a good for for AMT. See my comments below.

• The authors focus on statistical results of the comparison, especially the mean biases. But I don't think that mean biases could tell much about the uncertainties of TROPOMI and OMI. The standard deviation of the mean biases is large, which made me wonder if the overestimates of underestimates are broadly consistent. If not, presenting the mean biases here may not help understand the performance of satellite retrievals. For example, how well do these retrievals capture the spatial and temporal variability of NO2, HCHO, and FNR? And how the errors in satellite retrievals affect the interpretations of the ozone sensitivity?

In the updated manuscript we now include root mean squared error (RMSE) statistics to help demonstrate the uncertainty (this term is used throughout the updated manuscript to describe all unresolved errors beyond systematic biases such as random errors and relative biases) in both OMI and TROPOMI. We have also deemphasized the discussion about median bias in the updated manuscript in order to allow for more equal focus on systematic biases and uncertainty/unresolved errors in the retrievals.

In response to a comment by Reviewer #2 below, we address the capability of OMI and TROPOMI to capture the spatial variability of NO2, HCHO, and FNRs observed. As for the temporal variability, low earth orbit (LEO) satellites obtain, at best, a single snapshot per day, so we don't get much temporal information from these spaceborne systems. What we can demonstrate is the capability of the satellites to capture the inter-daily magnitude variability of NO2, HCHO, and FNRs observed by airborne spectrometers. To demonstrate this, we calculated daily mean tropospheric column quantities of NO2, HCHO, and FNRs from both satellites and airborne data for the entire LISTOS domain, and within 0.35 degrees of the NYC city center (identified as the emission source region), to calculate daily correlation statistics. The following text was added to Sect. 3.4.2 of the updated manuscript to summarize this evaluation and results "Given the limited spatiotemporal data coverage provided by the LISTOS campaign, a robust understanding of the temporal capabilities of OMI and TROPOMI to retrieve FNRs is not possible. LEO satellites obtain, at best, a single snapshot of both HCHO and NO2 each day, so one could only hope to obtain daily variability of FNRs from these spaceborne systems. To determine whether OMI and TROPOMI could capture the variability of the daily mean tropospheric column quantities of NO2, HCHO, and FNRs over the entire LISTOS domain from airborne data, we compared these daily mean values from NASA OMI, QA4ECV OMI, and TROPOMI to the airborne observations. For NASA OMI, daily

correlation ( $\mathbb{R}^2$ ) values were 0.85 (p = 0.001), 0.58 (p = 0.03), and 0.26 (p = 0.20) for NO2, HCHO, and FNRs, respectively. For QA4ECV OMI, daily correlation values were 0.85 (p = 0.001), 0.80 (p = 0.002), and 0.47 (p = 0.06) for NO2, HCHO, and FNRs, respectively. For TROPOMI, daily correlation values were 0.92 (p = <0.001), 0.85 (p = <0.001), and 0.41 (p = 0.03) for NO2, HCHO, and FNRs, respectively. All daily correlation statistics for HCHO and NO2 were significant to a 95% confidence interval and suggest that both OMI and TROPOMI can capture the overall interdaily magnitudes of FNR indicator species. However, only TROPOMI could observe the daily variability of domain-wide FNRs within a 95% confidence interval. This suggests that unresolved errors in either HCHO or NO2 retrievals (the analysis from this study suggests uncertainty in HCHO are driving FNR bias variability) from OMI, using both the NASA and QA4ECV algorithms, are too large to confidently capture the inter-daily variability in FNRs.

The same analysis was conducted for NASA and QA4ECV OMI except just for retrievals near the large anthropogenic source regions in NYC (within 0.35 degrees of the city center) where relative errors due to satellite retrievals for FNR calculations were the lowest (see Fig. 6). Daily correlation ( $R^2$ ) values for FNR retrievals near the source region of NYC for NASA OMI (0.13; pvalue = 0.39) were reduced compared to domain-wide means and QA4ECV OMI (0.66; p-value = 0.01) correlations were improved near the source region of NYC. Indicator species correlation values from NASA OMI were degraded compared to the domain-wide analysis suggesting that this satellite product may not be able to capture inter-daily variability of FNRs even in large source regions. However, this analysis suggests that QA4ECV OMI data has the capability to retrieve daily variability of FNRs in large emission regions such as NYC to a statistically significant level. Overall, TROPOMI retrievals at both fine and coarse spatial resolutions evaluated in this study are able to capture daily variability of tropospheric FNRs over the entire domain and emission source regions better compared to OMI products.".

To gather a more complete picture of the extent to which each satellite retrieval product lose spatial information (variance) compared to airborne data, we follow a recent algorithm named SpaTial Representation Error EstimaTor (STREET) (Souri, 2022) using NASA OMI and TROPOMI retrieval data. This method creates semivariograms determining the changes in spatial variability with distance for a defined variable (for this case HCHO and NO2 trace gas columns). The following description and results were added to Sect. 3.4.2 of the updated manuscript "To understand the extent to which OMI and TROPOMI retrieval products lose spatial information (variance) compared to airborne data during the LISTOS campaign, we applied the algorithm named SpaTial Representation Error EstimaTor (STREET) (Souri, 2022) using NASA OMI and TROPOMI retrieval data. This method creates semivariograms determining the changes in spatial variability with distance for a defined variable (for this case we used tropospheric column HCHO and NO2). The maximum variance at which the modeled semivariogram levels off is defined as a sill and data sets with larger sill values possesses richer spatial information. Figure S10 shows semivariograms, and the fitted stable Gaussian function described in Souri et al. (2022a), applied to TROPOMI and NASA OMI compared to airborne NO2 columns. Concerning the comparison of TROPOMI and airborne data at  $0.05^{\circ} \times 0.05^{\circ}$  resolution, we observe airborne semivariogram as high as  $20 \times 10^{15}$

molecules cm-2, a factor of two larger than what TROPOMI achieves. At a ~20 km length scale, TROPOMI can only observe ~40% of the airborne spatial variance, indicating that the spatial representation error in TROPOMI is ~60% at this scale. Similarly, NASA OMI fails to recreate >50% of the maximum variance observed in airborne data at  $0.15^{\circ} \times 0.15^{\circ}$  resolution. At ~20 km length scale, the spatial loss of OMI is >70%.

Figure S10 depicts the semivariograms and fitted exponential curves applied to TROPOMI and airborne HCHO columns. Immediately evident is that both semivariograms level off at longer distances compared to the analysis of NO2. This stems from the fact that HCHO columns tend to be spatially more homogeneous in the region of the LISTOS domain. For most length scales, TROPOMI can relatively well replicate the spatial variance observed in airborne data (~70%), which is explainable by the fact that HCHO concentrations are not highly heterogeneous in this region. We do not present the semivariogram for NASA OMI HCHO columns as the underlying unresolved biases in OMI are very large, introducing artifacts that cannot be solely attributable to unresolved spatial scales. Overall, TROPOMI and OMI capture spatial variance of NO2 similarly, TROPOMI performs slightly better; however, OMI is unable to capture the spatial variability of observed HCHO due to unresolved biases in this retrieval product. Since TROPOMI is able to capture the observed HCHO variability to retrieve FNR spatial variability compared to OMI products.".

As for the impact of satellite retrieval errors on the interpretation of O3 sensitivity, the recent study by Souri et al. (2022a) shows that satellite retrievals errors, in particular the unresolved bias in HCHO products, is the largest source of uncertainty in using satellite FNRs to investigate O3 sensitivity. Here we propagate the errors calculated from NASA OMI, QA4ECV OMI, and TROPOMI to FNR calculations during LISTOS using Eq. (15) from Souri et al. (2022a) and created maps of relative error shown in a new Fig. 6. The following text has been added as Sect. 3.4.1 of the updated manuscript "There are numerous sources of error when using satellite retrievals of tropospheric column HCHO and NO2 for investigating surface-level or PBL O3 production sensitivity regimes. The primary uncertainty sources are using indicator species to infer the complex chemistry driving O3 production and destruction, horizontal spatial representation error, uncertainty in converting tropospheric columns to PBL and surface-level values, and satellite retrieval unresolved biases (Souri et al., 2022a). As for the impact of satellite retrieval errors on the interpretation of O3 sensitivity, the recent study by Souri et al. (2022a) shows that satellite retrievals errors, in particular the unresolved error in HCHO products, are the largest source of uncertainty in using satellite FNRs to investigate O3 sensitivity. Here we propagate the uncertainty (RMSE) calculated from NASA OMI, QA4ECV OMI, and TROPOMI to FNR calculations during LISTOS 2018 using Eq. (15) from Souri et al. (2022a) and created maps of the relative error (see Fig. 6). From this figure it can be seen that satellite retrieval errors in HCHO and NO2 contribute significantly to satellite-derived FNR relative errors. In the largest NOx emission source regions of NYC, where combined column abundances of HCHO and NO2 are largest, is where the lowest relative errors of FNRs occur. For TROPOMI, which has the smallest values of uncertainty/RMSE compared to both NASA and QA4ECV OMI algorithms for HCHO and NO2, relative errors are as

low as ~40%. Away from the emission region of NYC these relative error values reach as high as ~80%. Similar patterns of relative error in FNRs from NASA and QA4ECV OMI retrievals are derived; however, the lowest relative error values over NYC are ~50% and reach values up to 100%. The largest relative errors are seen outside the source region of NYC in QA4ECV OMI retrievals due to having the largest uncertainty in HCHO and lower column abundances of this species in the rural regions of the domain. In addition to the fact that the less noisy retrievals from TROPOMI result in lower relative errors in FNR data, Fig. 6 further demonstrates the larger uncertainty in OMI as the relative error patterns are more heterogeneous. The spatial averaging of TROPOMI data results in the lowest relative errors of all four satellite products discussed in this study. TROPOMI at the coarser ( $0.15^{\circ} \times 0.15^{\circ}$ ) spatial resolution had relative errors as low as 35% and only increase to ~60% outside of the source location of NYC.".

• It's also not clear to me how the statistical results drawn from a single field campaign can be generalized to other regions or other time periods. I'd strongly recommend the authors go beyond the statistical comparison, and have a more thorough discussions about the sources of uncertainties, and the associated errors, and whether their conclusions can be generalized.

We agree with the review that more flight days during the campaign would be ideal. But this campaign provided a unique opportunity to use airborne remote-sensing observations of tropospheric column NO2 and HCHO to validate both OMI and TROPOMI coincidently (the overlap of both spaceborne sensors is novel). Also, the airborne sensors allowed for evaluation of OMI and TROPOMI over large areas which equates to having hundreds of ground-based systems for validation. While having long-term observations for robust validation of satellite sensors is ideal, this case study is unique in that it provides information about the performance of both OMI and TROPOMI over variable emission source regions (urban to rural) and scenes with differing physical characteristics (e.g., surface albedo, tropospheric compositions, clouds, etc.). This is now emphasized in Sect. 2.3 of the updated manuscript. Furthermore, to provide the reader information about the statistical significance of the satellite/airborne data comparison correlation values in Table 2 which are statistically significant to the 95% confidence level are identified in the updated manuscript.

Finally, in response to Reviewer #2, in addition to this comment, Sect. 3.4.3 of the updated manuscripts now discusses potential sources of systematic bias and uncertainty in OMI and TROPOMI HCHO and NO2 retrievals and how they impact satellite-derived FNR products. The text for Sect. 3.4.3 is as follows: "As demonstrated in this study, median biases of OMI and TROPOMI HCHO and NO2 retrievals tend to cancel out when calculating tropospheric column FNRs. Figures S4 and S5 show that the median bias spatial distribution of all satellite HCHO and NO2 retrievals are similar with a small low median bias in column abundances near the source region of NYC and high biases in the background regions. Table S1 shows that AMF calculations from NASA OMI, QA4ECV OMI, and TROPOMI use many of the same input data sets for geophysical variables (e.g., surface albedo, cloud fraction, cloud radiance, etc.) resulting in

[revised manuscript text omitted]

**Specific Comments:**

Abstract: The abstract is lengthy. I'd suggest the authors shorten the abstract to include only the core findings of this work. For example, the first paragraph may belong to introduction.

The abstract has been shortened as much as possible.

Line 370: What are the quality flags for? Is this the same quality flag as for TROPOMI? If so, why do you choose different thresholds? Better to include references here.

OMI data user's manuals for NO2 and HCHO state the in order to use the highest quality data that only pixels with qa\_values = 0. For TROPOMI, the individual species data user's manuals for NO2 and HCHO make recommendations for the qa\_values used in this study (0.75 for NO2 and 0.5 for HCHO) in order to use high quality data. The HCHO data user's manuals for TROPOMI recommends removing data with qa\_values < 0.5 which we followed in the original manuscript. We tested whether using the higher qa\_value of 0.75 to filter TROPOMI HCHO retrievals would impact the results of this study. When removing TROPOMI HCHO pixels with qa\_values < 0.75 the statistics had a very minor change and the results of the study remained consistent. In the updated manuscript we have added the following clarifications in Sect. 2.5: "Satellite retrievals with high quality were filtered for use by removing individual retrievals that did not have quality flags (qa) = 0 for HCHO and NO2 when applying OMI data. This qa value is suggested in the OMI data user's manuals for the application of the highest quality science data and for the removal of OMI pixels impacted by the row anomaly. For TROPOMI, individual retrievals of NO2 and HCHO that had qa < 0.75 and qa < 0.5 were removed prior to spatial averaging, respectively, as recommended by the TROPOMI data user manuals for each species.".

Table 2: I'd suggest include an estimate of the error, such as normalized mean standard errors. NMB doesn't tell much about the precision of the retrievals.

We have added root mean squared error statistics throughout the updated manuscript.

Line 530: Maybe you could have a figure of the mean biases of HCHO to show where OMI or TROPOMI HCHO is biased high?

Supplemental Fig. S4 and S5 in the updated manuscript now show the spatial distribution of campaign-averaged OMI and TROPOMI NO2 and HCHO biases during LISTOS.

Line 600: I'm not sure if we could call this as a 'high pollution' day because ozone was actually low on this day. This could very well be a cold day when the lifetime of NO2 is long, and the the photolysis is low. I'm not sure how much value there is to evaluate FNR on this day. It'd be more interesting to add another day with both high ozone and high NO2.

In response to a comment by Reviewer #2, and to reduce the length and complexity/density of the manuscript, we have removed this section from the updated paper version as it did not add much value to the overall study.

Line 700: It is interesting to see that improved the a priori from CMAQ does not improve the retrieval performance of OMI. The authors attribute this to coarse resolution of OMI. Could this be due to the coarse resolution of cloud and surface albedo data used in the retrieval?

We attribute the degradation in OMI retrieval performance when using high spatial resolution CMAQ data for a priori data primarily due to the too steep shape profile in the model data. This was explained in detail in the original manuscript. Overall, for tropospheric NO2 retrievals from OMI, the shape factor from CMAQ has higher NO2 concentrations in the PBL and lower values in the free troposphere compared to the a priori profiles used in standard NASA OMI retrievals resulting in lower air mass factors (AMF). As discussed in this manuscript, and in other papers referenced in the manuscript (e.g., Goldberg et al., 2017), the differences in shape factors produced by CMAQ in comparison to the coarser global model output use as a priori information in OMI is the primary reason for the poorer performance in NO2 retrievals. We compared the "raw" WRF-CMAQ output to airborne observations and confirmed the model biases and "scaled" the higher spatial resolution model data to better replicate observations. While median biases of OMI NO2 and HCHO retrievals reprocessed with the scaled WRF-CMAQ a priori information was only moderately improved compared to the standard retrievals, the spatial variability of the species was better retrieved compared to observations. The following text updates were added to Sect. 3.3 of the updated manuscript to reflect this point: "Scaled NASA OMI tropospheric column NO2 and HCHO retrievals had smaller median biases of  $-0.3\pm3.9 \times 10^{15}$  molecules cm-2 and  $4.4\pm7.1 \times 10^{15}$ molecules cm-2 and much lower RMSE values of  $3.9 \times 10^{15}$  molecules cm-2 and  $7.8 \times 10^{15}$ molecules cm-2, respectively, compared to the retrievals with raw WRF-CMAQ predictions. This result demonstrates the need for accurate shape factors (i.e., vertical distribution of trace gases) to be used as a priori information in NASA OMI retrievals. Finally, the improved accuracy of tropospheric column NO2 and HCHO retrievals using scaled WRF-CMAQ model predictions resulted in a slightly higher magnitude of FNR median bias  $(0.5\pm3.2)$ ; however, with lower RMSE values, compared to reprocessed data using raw CMAQ predictions. In comparison to standard NASA OMI products, the reprocessed satellite data using scaled WRF-CMAQ a priori information had similar median biases in FNR values and lower median biases for HCHO  $(4.4\pm7.1\times10^{15}$

molecules cm-2) and NO2 (- $0.3\pm3.9 \times 10^{15}$  molecules cm-2). All reprocessed data variables using scaled model simulated shape factors, due to the reduction in uncertainty in retrieve HCHO and NO2 data, had lower RMSE values, higher correlation (except for FNR), and similar to better linear regression slopes compared standard satellite retrievals."

A detailed discussion of how the coarse spatial resolution input geophysical data sets (e.g., cloud fraction/radiation, surface reflectance, etc.) used in OMI and TROPOMI, can contribute to systematic bias and uncertainty in the FNR retrievals is now provided in Sect. 3.4 of the updated manuscript.

Line 835: While the low mean biases of FNR is low, the standard deviation is very large. The R sure is also low for FNR. Thus I don't think the errors of HCHO and NO2 could cancel out. The errors in HCHO and NO2 can offset only if the errors are correlated. I'd suggest the authors make a scatter plot for errors of HCHO versus NO2, and see if they are correlated.

The manuscript has been re-written in a way that it is clear that the systematic/median biases cancel out in FNR calculations. However, as identified by the reviewer, the uncertainty in HCHO and NO2 retrievals do not cancel out in FNR calculations. Biases for HCHO and NO2 retrievals from NASA OMI and TROPOMI are not correlated with  $R^2$  values <0.05. This is described in detail in the updated manuscript in Sect. 3.4.

---

## Author Comment (AC2)

**Response to Reviewer #2**

We thank Reviewer #2 for reviewing the manuscript and for their helpful comments. We feel that the changes suggested have resulted in improvements in the quality of the study. All reviewer comments are in italics and the author's responses are in standard font.

*Johnson et al. present a detailed statistical analysis of FNR observations with two different OMI products, and with TROPOMI. FNR, NO2 and HCHO satellite retrievals are validated against airborne measurements over the New-York area during summer 2018. It is well demonstrated in the paper that the noise of the HCHO satellite retrievals is the limiting factor for the FNR observations since the individual HCHO columns need to be averaged at poorer time and space resolution than NO2. The precision of the OMI HCHO observations does not allow for daily FNR observations at OMI native resolution. The OMI QA4ECV HCHO product is found to perform better than the OMI NASA HCHO product. TROPOMI offers an important improvement in the spatial and temporal resolution of HCHO and NO2 tropospheric columns, allowing for daily FNRs retrievals at TROPOMI native spatial resolution. The results are further improved by averaging TROPOMI observations on a larger spatial grid. Both NO2 and HCHO satellite products suffer from bias compared to aircraft observations. This study identifies an important positive bias over rural regions (lowest columns) for both species, and for OMI and TROPOMI products. However, the positive bias found for the TROPOMI products is reduced compared to OMI thanks to the better spatial resolution and lower noise. It is also demonstrated that the bias of the FNR satellite observations is much lower than the respective NO2 and HCHO biases. This is an important result that would deserve more discussion in the paper. The paper is well written, albeit a bit long and too detailed. The scientific approach is solid, however some points should be tested or clarified. I recommend publication in AMT after some revisions.*

*General comments*

*One concern is the small number of days that are available for the validation. Here the field campaign covers only a few days (8 days collocated with OMI, 12 days with TROPOMI). The statistical results are not always significant, especially for OMI. Studies on longer time period could improve the observed correlations, that are poor for HCHO.*

We agree with the review that more flight days during the campaign would have been ideal. However, the LISTOS field campaign provided a unique opportunity to use airborne remote-sensing observations of tropospheric column NO₂ and HCHO to validate both OMI and TROPOMI coincidently (the overlap of both spaceborne sensors is novel). Also, the airborne sensors allowed for the evaluation of OMI and TROPOMI over large areas which equates to having hundreds of ground-based systems for validation. While having long-term observations for robust validation of satellite sensors is ideal, this case study is unique in that it provides information about the performance of both OMI and TROPOMI over variable emission source regions (urban to rural) and scenes with differing physical characteristics (e.g., surface albedo, tropospheric compositions,

etc.). This is now emphasized in Sect. 2.3 of the updated manuscript. Furthermore, to provide the reader information about the statistical significance of the satellite/airborne data comparison correlation values in Table 2 which are statistically significant to the 95% confidence level are identified in the updated manuscript.

• *The paper could be improved by providing an information about the spatial and temporal resolution that might provide useful FNR observations with OMI (ex. monthly averaged data). How many observations are needed at minimum to reduce the noise at the level of the TROPOMI daily observations?*

We agree with the reviewer that this is an interesting aspect of applying satellite data to derive FNR data for $O_3$ production sensitivity analysis. See a very similar response to Reviewer #1 for their comment about spatiotemporal resolution needed for trend studies. In response to both reviewers, we added an entire section (Sect. 3.4.2) in the updated manuscript which describes the capabilities of OMI and TROPOMI to observe spatial and temporal variability of FNRs during LISTOS.

[revised manuscript text omitted]

Figure S10 depicts the semivariograms and fitted exponential curves applied to TROPOMI and airborne HCHO columns. Immediately evident is that both semivariograms level off at longer distances compared to the analysis of $NO_2$. This stems from the fact that HCHO columns tend to be spatially more homogeneous in the region of the LISTOS domain. For most length scales, TROPOMI can relatively well replicate the spatial variance observed in airborne data (~70%), which is explainable by the fact that HCHO concentrations are not highly heterogeneous in this region. We do not present the semivariogram for NASA OMI HCHO columns as the underlying unresolved biases in OMI are very large, introducing artifacts that cannot be solely attributable to unresolved spatial scales. Overall, TROPOMI and OMI capture spatial variance of $NO_2$ similarly, TROPOMI performs slightly better; however, OMI is unable to capture the spatial variability of observed HCHO due to unresolved biases in this retrieval product. Since TROPOMI is able to

capture the observed HCHO variability to a sufficient degree, combing these two facts suggest that TROPOMI has better capability to retrieve FNR spatial variability compared to OMI products.".

*• It would be interesting to know if the HCHO observations with aircraft instruments are also noisier than the NO2 observations, and therefore also the limiting factor of suborbital FNR observations.*

It is expected that HCHO retrievals will be nosier compared to $NO_2$. There are two primarily reasons for this: 1) optical depths for HCHO peak in the UV range (<380 nm) at the same wavelengths coinciding with large Rayleigh scattering and optical depths of ozone leading to a weak/noisy signal, and 2) the stronger $NO_2$ optical depths in the visible wavelength range (400-500 nm), where there are higher signal-to-noise ratios, permits retrievals with less noise. Nowlan et al. (2018) derived the precision of the GCAS/GeoTASO airborne remote-sensing systems used for $NO_2$ and HCHO retrievals in this study. Nowlan et al. (2018) quantified precisions of $1.0 \times 10^{15}$ molecules cm$^{-2}$ and $1.9 \times 10^{16}$ molecules cm$^{-2}$ at a fine spatial resolution of 250 m $\times$ 500 m for $NO_2$ and HCHO, respectively. Averaging these precision values to the spatial resolution of $0.05° \times 0.05°$ improves these precision levels to $6.4 \times 10^{13}$ molecules cm$^{-2}$ and $1.2 \times 10^{15}$ molecules cm$^{-2}$ for $NO_2$ and HCHO, respectively. The campaign-averaged column $NO_2$ and HCHO abundances from GCAS/GeoTASO at $0.05° \times 0.05°$ were $6.6 \times 10^{15}$ molecules cm$^{-2}$ and $1.5 \times 10^{16}$ molecules cm$^{-2}$, respectively. Comparing the precision values of Nowlan et al. (2018) to the mean abundances during LISTOS at the same spatial resolution results in mean precision levels of 1% and 8% for $NO_2$ and HCHO, respectively. Overall, the HCHO airborne data for is expected to have a factor of 5-10 more noise compared to $NO_2$. Text describing this have been added to Sect. 3.2.6 of the updated manuscript.

The reason why we resort to using precision statistics from Nowlan et al. (2018), and not the LISTOS data set, is that airborne flight tracks during LISTOS were focused on the source region of NYC, and surrounding areas, which did not allow us to define a "clean" region for both $NO_2$ and HCHO. A caveat to using precision statistics from Nowlan et al. (2018) is that the observations were obtained at different locations/times and under different atmospheric and viewing geometry conditions which could results in different signals. However, we feel that the large difference in noise derived in the manner explained above is sufficient to assume that the airborne HCHO retrievals are nosier compared to $NO_2$.

*• In Table 2, I recommend adding a line providing the mean value +- the standard deviation of FNR, HCHO and NO2 for the aircraft, NASA OMI, QA4ECV OMI, and TROPOMI (0.15° and 0.05°).*

This information has been added to Table 2 of the updated manuscript.

*• I recommend more tests on the selection of the data, that is currently at the edge of the statistical significance (see later).*

*One interesting result of the paper is that the errors in NO2 and HCHO columns tend to offset in the FNR observations. There might be good reasons for this, such as error cancellation. It is*

*therefore important to use HCHO and NO2 products that have been retrieved with algorithms and auxiliary data as consistent as possible. This is an important message for the future TEMPO product.*

In response to this comment and Reviewer #1, the manuscript has been re-written in a way that it is clear that the systematic/median biases of HCHO and $NO_2$ retrievals tend to cancel out in FNR calculations. However, the uncertainty in HCHO and $NO_2$ retrievals when compared to airborne observations do not cancel out. This is clear as the unresolved error/RMSE values for FNRs are still large. Furthermore, biases for HCHO and $NO_2$ retrievals from NASA OMI and TROPOMI are not correlated with $R^2$ values <0.05. This is now described in detail in the updated manuscript.

Below, in response to your next comment, and in the updated manuscript, we discuss in detail about potential reasons why median errors tend to cancel out while unresolved errors do not.

*• It would be good to discuss further what type of error might cancel out, or at least might reduce, when using NO2 and HCHO retrieved using consistent algorithms to derive FNR (surface albedo, cloud products, a priori profiles).*

In direct response to this comment a discussion section (Sect. 3.4.3) of the updated manuscript has been added. Furthermore, two additional sections have been added (Sect. 3.4.1 and 3.4.2) to discuss the spatial and temporal capabilities of OMI and TROPOMI as well as relative errors of these satellite retrievals. The text for Sect. 3.4.3 is as follows: "As demonstrated in this study, median biases of OMI and TROPOMI HCHO and $NO_2$ retrievals tend to cancel out when calculating tropospheric column FNRs. Figures S4 and S5 show that the median bias spatial distribution of all satellite HCHO and $NO_2$ retrievals are similar with a small low median bias in column abundances near the source region of NYC and high biases in the background regions. Table S1 shows that AMF calculations from NASA OMI, QA4ECV OMI, and TROPOMI use many of the same input data sets for geophysical variables (e.g., surface albedo, cloud fraction, cloud radiance, etc.) resulting in campaign-averaged AMFs of HCHO, $NO_2$, and the ratios of these products (AMF FNRs) which are relatively similar across the LISTOS domain (see Fig. S11). For all satellite products, HCHO and $NO_2$ AMFs have much less variability compared to AMFs derived for airborne data which along with SCD biases may contribute to the median high biases in background HCHO and $NO_2$ retrievals. A primary reason for the inability of satellites to capture AMF variability over the LISTOS domain is likely the shape factors being used for these calculations having spatial resolutions of $1.0° \times 1.0°$ to even coarser grids (Table S1). Furthermore, while TROPOMI and QA4ECV OMI retrievals used daily model data for shape factor calculations, NASA OMI uses monthly products which will be challenged to capture the large spatiotemporal variability of tropospheric HCHO and $NO_2$ vertical profiles in urban and rural regions occurring in reality. Finally, coarse geophysical input data sets used in AMF calculations (see Table S1) will not capture the spatial distribution of these variables in reality. Airborne AMF calculations use much higher spatial resolution input data sets (e.g., 500 m surface albedo data (Judd et al., 2020) compared to $0.5° \times 0.5°$ or coarser surface reflectivity products used in OM and

TROPOMI) and shape factors are calculated with 12 km × 12 km CMAQ model simulations which both aid in the much larger spatial variability of AMFs not captured in satellite retrievals.

The more interesting aspect found in this study is that unresolved errors in HCHO and $NO_2$ retrievals don't cancel out in FNR calculations as do the systematic/median biases. While there are some reasons why uncertainty in HCHO and $NO_2$ retrievals could stem from opposite impacts of geophysical parameters in AMF calculations, such as AMF uncertainties in HCHO and $NO_2$ having opposite trends with increasing surface reflectance (comparing Fig. 10 from De Smedt et al. (2018) and Fig. 20 from Liu et al. (2021)), these differences are minor and overall AMF calculations for both species in NASA OMI, and QA4ECV OMI, and TROPOMI have similar input data sets. A portion of the uncertainty of HCHO and $NO_2$ retrievals not canceling out stems from the AMF calculations shown in Fig. S11. In order for HCHO and $NO_2$ AMFs to have no impact on VCD uncertainty cancelations, AMF FNRs would be a constant or similar value at all locations. However, from Fig. S11 it is shown that AMF FNRs, while having smooth spatial variability, are not a constant value. Therefore, some of the unresolved error residual in the FNR calculations will be due to differences in HCHO and $NO_2$ AMF calculations. This is emphasized in NASA OMI AMF FNR plots in Fig. S11 where different CTMs, at different spatial resolutions (see Table S1), are used to derive HCHO and $NO_2$ shape factors leading to noticeable differences in the respective AMF calculations. This likely is one of the reasons that NASA OMI FNRs have the largest uncertainty (highest bias standard deviation and RMSE values) compared to airborne data (see Table 2) of all OMI and TROPOMI satellite products. Finally, the airborne AMFs are more variable compared to satellite products due to the finer-scale shape factors and geophysical parameter input data used in AMF calculations which satellites inherently are not able to capture, contributing to the satellite uncertainty.

The rest of the remaining unresolved error in FNR calculations is likely due to the SCD retrievals from OMI and TROPOMI sensors. As demonstrated in this study the uncertainty in both OMI and TROPOMI retrievals of HCHO is large. The SCD retrievals of HCHO from TROPOMI have been shown in the past to have less noise compared to OMI due to the higher spatial resolution and at least the same signal-to-noise (De Smedt et al., 2021). The larger uncertainty in OMI retrievals of HCHO compared to TROPOMI directly leads to the higher bias standard deviation and RMSE values for derived FNRs in OMI compared to TROPOMI (see Table 2). This is further emphasized in the spatially-averaged TROPOMI data (at 0.15° × 0.15° to match OMI data) where HCHO and FNR retrievals have a factor of 2-3 lower RMSE compared to NASA OMI and QA4ECV OMI. TROPOMI $NO_2$ SCDs have also been shown to have less noise compared to OMI retrievals due to the higher spatial resolution and similar signal-to-noise (van Geffen et al., 2020, 2022). This is also shown in Table 2 when averaging TROPOMI data to match the OMI spatial resolution. Overall, HCHO and $NO_2$ SCD noise contributes to uncertainty in OMI and TROPOMI VCDs and are not cancelled out in FNR calculations; however, the reduced noise in TROPOMI SCD retrievals leads to improved VCDs of HCHO and $NO_2$ abundances and the ratios of these products.".

*• I recommend adding a table providing a quick look at the auxiliary data used in the AMF calculations for the NASA, QA4ECV and TROPOMI products, and TEMPO.*

This has been added as Supplemental Table S1 in the updated manuscript.

*• Discuss the different FNR biases with the level of consistency between NO2 and HCHO AMF settings.*

Please see the discussion above, and new Sect. 3.4.3 in the updated manuscript, which addresses this comment.

*The low HCHO correlations are also partly due to lower spatial variability of the HCHO distribution compared to NO2, also in the airborne measurements, over the time and domain of the study.*

We agree with the reviewer that the spatial variability of HCHO is lower compared to $NO_2$ during the study. However, we feel that the low correlation of the satellite/airborne tropospheric HCHO data is primarily due to the inability of the satellites to capture the spatial variability of observed HCHO.

*Selection of data:*

*• Filter row anomaly both for HCHO and NO2 products.*

The pixels impacted by the row anomaly were removed using data quality flags in both OMI HCHO and $NO_2$.

*• The lower bound limits for HCHO and NO2 appear to be strict, compared to the reported standard deviations of the bias. For HCHO, the bias std ranges from 9 to 5e15 molec.cm-2, while the lower limit has been set to -8e15. For NO2, bias std is about 4e15, while the lower limit has been set to -1e15 molec.cm-2. There is a possibility that a significant part of the negative values has been filtered out while it actually belongs to the normal distribution. The effect could be an artificial increase of the mean background values. Please test a lower bound limit for the data selection.*

The lower and upper bounds were based on suggested limits used in recent OMI and TROPOMI validation studies (e.g., Zhu et al., 2020) and personal communication with OMI $NO_2$ retrieval team. However, to test whether the lower limit of HCHO and OMI impacted the high bias in background concentrations retrieved in OMI and TROPOMI, we reduced the lower bound of HCHO and $NO_2$ to $-5.0 \times 10^{16}$ molecules cm$^{-2}$ $-1.0 \times 10^{16}$ molecules cm$^{-2}$, respectively. The statistical comparison during LISTOS was not impacted by reducing this lower limit.

*• At the spatio-temporal resolution of the study, OMI retrievals are clearly at their detection limit. Please consider testing a lower grid resolution (0.2°) for OMI.*

As explained in Sect. 2.6 of the original manuscript, now Sect. 2.5 of the updated version, we apply a point oversampling technique when spatially averaging the retrievals. When averaging OMI data to the $0.15° \times 0.15°$ spatial resolution (standard radius of $0.075°$), we employed a radius twice the

standard size equal to $0.15°$. This helps avoid issues due to the fact that the $0.15° \times 0.15°$ grids are near the native spatial resolution of OMI at nadir.

*• To increase the number of collocations, I would suggest testing a larger temporal window of 3h for the airborne retrievals.*

Increasing the temporal threshold to lengths greater than 1 hour will increase temporal data representativity error. We have already adopted a longer temporal threshold compared to other satellite validation studies using GCAS/GeoTASO observations during LISTOS (e.g., Judd et al., 2020). This issue is particularly true for $NO_2$ near the surface where it's lifetime can be minutes to hours. However, to test how increasing the temporal colocation threshold to 3 hours would impact the statistics we conducted this sensitivity test for NASA OMI. As expected by the reviewer, this increased satellite/airborne colocations by ~70%. However, it degraded the statistical evaluation of the satellite retrievals especially for $NO_2$ where median biases, bias standard deviations, correlation, and RMSE were noticeable worse compared to using a temporal colocation threshold of 1 hour. Given that correlation statistics are mostly significant to a 95% confidence interval using the limited number of colocations (see our response above) using the threshold of 1 hour, and we already use a temporal threshold longer than others evaluating satellites with GCAS/GeoTASO observations, we kept our statistical analysis using the threshold of 1 hour for our updated manuscript.

*It would be good to better stress the specificities of this paper compared to the recent paper of Souri et al., 2022, which also compares OMI and TROPOMI NO2, HCHO and FNR errors over the US. (Characterization of Errors in Satellite-based HCHO / NO2 Tropospheric Column Ratios with Respect to Chemistry, Column to PBL Translation, Spatial Representation, and Retrieval Uncertainties)*

The recent paper by Souri et al. (2022a) assessed the major error components of retrieving FNRs using satellite data. The primary driver of uncertainty in satellite-derived FNRs identified in this study was from systematic bias and unresolved error of the $NO_2$ and HCHO retrievals themselves (HCHO retrieval uncertainty being the main issue). Souri et al. (2022a) estimated TROPOMI biases using stationary point-source observation data (MAX-DOAS) and OMI errors using airborne in situ data. This study builds off these findings to better characterize OMI and TROPOMI FNR retrieval error using a unique validation data set (i.e., GCAS and GeoTASO) providing coincident $NO_2$ and HCHO information obtained during the LISTOS field campaign. This particular data set has not yet been used to assess HCHO, $NO_2$, and resulting FNR retrieval errors. As explained in the response above to the reviewer comment about the choice of using LISTOS campaign data, this unique data set provides information about the performance of coincident HCHO and $NO_2$ retrieval from both OMI and TROPOMI over variable emission source regions (urban to rural) and scenes with differing physical characteristics (e.g., surface albedo, tropospheric compositions, clouds, etc.). This is emphasized in the updated manuscript.

*Detailed comments*

*Abstract*

*Line25: "high spatiotemporal coverage": please provide numbers, such as the native resolution of OMI and TROPOMI. I would rephrase "OMI and TROPOMI are capable of providing NO2 and HCHO daily global observation at native resolution of respectively ... and ...". However, satellite observations are known to be affected by noise and biases, that limit the precision of FNR.*

In order to shorten the abstract, in response to Reviewer #1, we have removed much of this discussion as we provided these details in the main body of the text.

*Line 25: "..., yet a recent study suggested ....". This sentence is rather vague. Which study?*

This statement has been removed from the abstract.

*Line 30: Please specify the covered period.*

This has been added to the abstract.

*Line 32: Please be clearer in the abstract with the term "suborbital". This is not obvious for a general reader.*

This has been replaced with "aircraft-based".

*Line 49: Place replace large by larger biases.*

Corrected.

*Introduction*

*Line 95: please add the 2 following references: Wang et al, 2022; Harkey et al., 2015.*

These references don't use both satellite HCHO and $NO_2$ to study ozone production sensitivities so would not be appropriate to cite here.

*Line 100: the choice of references seems weird. It might be good to add references for NO2 and HCHO L2 products of each sensor, and not only for studies using both species together. The SCIAMACHY instrument is missing in the list.*

This sentence has been updated to read "Multiple past and current space-based spectrometers have the capability to retrieve simultaneous $NO_2$ and HCHO tropospheric columns to calculate FNRs for studying $O_3$ production sensitivity regimes including…" in order to emphasize the purpose of this statement. The purpose of this statement is to identify spaceborne systems which have been used for studying $O_3$ production sensitivity regimes and is why we chose the specific references. As requested by the reviewer we have added SCIAMACHY to this sentence.

*Methods*

*Line 163: The OMI rows affected by the row anomaly should be filtered out in the HCHO product such as in the NO2 product. The reference sector method does not correct for the row anomaly, but for the stripes between the valid rows. Please rephrase (and check that the HCHO data are filtered correctly).*

The pixels impacted by the row anomaly were removed using data quality flags in both OMI HCHO and NO$_2$. The text in the updated manuscript is now clearer in this section: "The row anomaly in NO$_2$ and HCHO retrievals was avoided in this study using data quality flags to filter out rows/pixels flagged by the row anomaly detection algorithm."

*Line 204: Please explain what you mean by "iterative fitting algorithm" and "simultaneous fitting". To me, a DOAS fit is an iterative fit (least-squared fit).*

We thank the reviewer for identifying our misinterpretation of the literature. Given this statement was not necessary for the study, it has been removed in the updated manuscript.

*Line 215: The QA4ECV fitting window is 328.5-359 nm, such as TROPOMI. For all HCHO products, please double check the retrieval intervals that are mentioned in the paper. Most of the recent retrievals use a fitting window larger than 328.5-346 nm.*

We thank the reviewer for identifying this error in the text. The fitting window ranges have been corrected for each sensor/algorithm.

*Line 261: Please explicit the term SWs.*

This sentence has been removed as described above.

*Results*

*Line 444: "Tropospheric columns NO2 concentrations", "tropospheric columns NO2 retrievals". Could be simplified to "Tropospheric NO2 columns" and homogenized throughout the paper.*

These phrases for tropospheric column NO$_2$ and HCHO have been simplified as suggested by the reviewer.

*Line 465: It should be emphasized here that TROPOMI offset for low columns is lower than OMI at the resolution of 0.05.*

The following sentence was added to the updated manuscript: "TROPOMI at its near native spatial resolution has the least high bias of background tropospheric NO$_2$ columns demonstrated by the lower y-axis intercept compared to all OMI and TROPOMI data products at the coarser spatial resolution".

*Line 491: add a reference to Verhoelst et al. 2021.*

Added.

*To our knowledge, the cited references do not report a high bias of NO2 for background values. But the studies were made with the previous version of the TROPOMI NO2 product. This should be clarified here.*

We agree with the reviewer and the sentence has been correct to read: "The results here suggest that OMI, and to a lesser extent TROPOMI, tropospheric column NO$_2$ retrievals errors have a magnitude dependence and tend to have some high bias in rural/background regions and a low bias in moderately to highly polluted regions which agrees with past validation studies (e.g., Zhao et

al., 2020; Lamsal et al., 2021; Verhoelst et al., 2021).". OMI has been shown to have a high bias in clean regions (e.g., Lamsal et al., 2021) which are larger compared to TROPOMI (Zhao et al., 2021). Many studies show that OMI and TROPOMI $NO_2$ data compare well to stations located in clean/background sites; however, this study applying airborne remote-sensing data is better able to retrieve clean and polluted regions in the same location on the same day compared to previous validation sites.

*Line 495. The comparison of TROPOMI NO2 Bias at 0.05 and 0.15° also clearly shows the spatial resolution effect on the background values (from negative to positive and similar to OMI NMB). Please mention this resolution effect.*

The following sentence has been added: "Finally, TROPOMI $NO_2$ data averaged to the coarser spatial resolution of OMI has a similar campaign-averaged high median bias as both OMI retrieval algorithms; however, displayed RMSE values nearly twice as small as NASA and QA4ECV OMI, further emphasizing the importance of spatial resolution for retrieving tropospheric $NO_2$ columns.".

*Table2: Please add one line with the mean FNR, NO2 and HCHO columns and their standard deviations.*

This information has been added to Table 2.

*Figure 3: Please test different data selection as suggested in the general comments.*

We tested the lower limit of $NO_2$ and HCHO values, increased qa_values for TROPOMI HCHO, and increased temporal colocation threshold as suggested by the reviewer (described above). Decreasing the lower limit of $NO_2$ and HCHO values from OMI and TROPOMI and increasing the qa_value for filtering TROPOMI HCHO had no impact on the statistical results of the study and were kept the same in the updated manuscript due to selecting these values based on satellite data user's manuals, past validation studies, and personal communication with algorithm teams. Increasing the temporal colocation threshold to 3 hours increased the number of colocations for statistical evaluation; however, also increased spatial representation errors which degraded the statistics of the satellite retrievals (especially for $NO_2$ which has a shorter atmospheric lifetime compared to HCHO). These suggestions were good for testing the robustness of our satellite evaluation methods; however, for the reasons above were not included in the updated manuscript.

*Line 517: The results are not so much in agreement with the study of Vigouroux, who reported indeed a high bias for the lowest columns, but for columns lower than 2.5e15 molec.cm-2. The TROPOMI bias ranges from 0 to negative values for columns larger than 5e15 molec.cm-2.*

See our response to the similar reviewer comment below.

*Line 527: Please also compare the bias standard deviation between OMI and TROPOMI.*

More emphasis on discussing bias variability and uncertainty using RMSE statistics for all retrievals has been added to the updated manuscript.

*Line 530: In De Smedt 2021, it is reported that the OMI HCHO offset is larger than for TROPOMI. But the reported bias are all negative for columns larger than 5e15 molec.cm-2. The conclusions of this study are therefore not completely in agreement with De Smedt et al. or with Vigouroux et al..*

In order to provide a more quantitative comparison with the recent validation studies of OMI and TROPOMI HCHO (Vigouroux et al., 2020; De Smedt et al., 2021), we separated our collocated satellite/airborne data points using clean ($<5.0\times10^{15}$ molecules cm$^{-2}$) and polluted ($\geq8.0\times10^{15}$ molecules cm$^{-2}$). We chose a slightly higher threshold for separating clean HCHO columns to optimize the number of colocations for statistics and to be as similar as possible to Vigouroux et al. (2020). We also added a highly polluted threshold ($>16.0\times10^{15}$ molecules cm$^{-2}$) to further emphasize our results. The table below summarizes the median bias ± bias standard deviation and NMB results for NASA OMI, QA4ECV OMI, and TROPOMI at coarser/fine spatial resolution for the different HCHO column magnitudes.

**Statistical evaluation of NASA OMI, QA4ECV, and TROPOMI retrievals of tropospheric column HCHO. Statistics presented are median bias ± bias standard deviation and NMB (%).**

| | NASA OMI (0.15° × 0.15°) | | | | QA4ECV (0.15° × 0.15°) | | |
|---|---|---|---|---|---|---|---|
| | Clean | Polluted | Highly Polluted | | Clean | Polluted | Highly Polluted |
| Bias | 2.8±6.2 | 4.6±7.9 | -2.3±9.2 | Bias | 2.7±7.3 | 2.1±8.7 | -3.8±7.4 |
| NMB | 75.1 | 30.3 | -8.9 | NMB | 72.1 | 13.7 | -14.6 |
| | TROPOMI (0.15° × 0.15°) | | | | TROPOMI (0.05° × 0.05°) | | |
| | Clean | Polluted | Highly Polluted | | Clean | Polluted | Highly Polluted |
| Bias | 3.1±1.4 | 1.8±4.4 | -2.2±4.8 | Bias | 2.4±2.3 | 1.3±6.5 | -2.7±7.0 |
| NMB | 78.1 | 12.5 | -8.7 | NMB | 60.9 | 8.5 | -10.1 |

While the positive tropospheric HCHO column biases derived in our study are higher compared to the recent studies of Vigouroux et al. (2020) and De Smedt et al. (2021), the magnitude dependance is similar. We show here that clean/background satellite HCHO columns are larger than observations for all satellite products and transition to a low bias in highly polluted regions. Text describing this, and the table above was added as Table S3, was additional text was added to the updated manuscript in Sect. 3.2.3.

*Line 594: I agree with the reasons for the poor HCHO correlation. Please add that they are also partly due to the low HCHO variability over the studied time and domain. A full year study would result in larger correlations.*

We agree with the reviewer that the low correlations between the satellite and observed HCHO is primarily driven by the spatial variability in this study. This differs from many recent studies which use stationary point-source observations (Vigouroux et al., 2020; De Smedt et al., 2021) which primarily capture temporal variability in column HCHO retrievals. This may suggest that temporal

variability in HCHO is easier to retrieve from space compared to spatial variations which rely on input geophysical and a priori data sets (e.g., surface albedo, aerosol, a priori profiles, clouds) to accurately capture entire scenes variability of the specific variable. This has been expanded on in the updated manuscript.

*High pollution case study: The added value of this section is not clear. As the paper is already long and detailed, I would suggest removing this section. If not removed, I then suggest to discuss the causes of higher NO2 columns and lower HCHO columns, such as surface temperature.*

We agree with the reviewer and this section has been removed.

*Common a priori sensitivity tests:*

*- It is not clear why the WRF-CMAQ profiles need to be scaled for the NASA OMI datasets, but not for the TROPOMI datasets.*

The primary reason for differences between OMI and TROPOMI retrievals using the WRF-CMAQ a priori profiles is the difference between the shape factors derived from WRF-CMAQ and the a priori information used in OMI (GMI) and TROPOMI (TM5). The exact comparison of the shape factors produced by WRF-CMAQ, GMI, and TM5 is inhibited by the fact that the standard retrieval products of tropospheric $NO_2$ from OMI and TROPOMI do not provide this a priori profile information. Therefore, this hypothesis could not be tested in this study. The impact of higher spatial resolution model simulations when used as a priori information was described in the original version of the manuscript and compared to other studies seeing the same results.

*- Figure 6: please explain in the legend what is the NASA OMI (scaled).*

The figure caption explains this in the original manuscript. We have slightly updated it to now read: "The OMI FNR retrievals calculated with the scaled WRF-CMAQ profiles are identified as "scaled" in the figure panel titles.".

*- Comparing Table 2 and Table 4, I can only see an improvement for TROPOMI at 0.05° resolution. The added value of this section is not clear, given the uncertainties in the WRF-CMAQ profiles.*

It should be noted when comparing Table 2 and Table 3 in the updated manuscript that various aspects of OMI retrievals were also improved when using the WRF-CMAQ shape factors for AMF calculations. This section has been rewritten to better emphasize aspects of the retrievals that were improved by the higher spatial resolution model a priori profiles. We wanted to include this section of the manuscript as there is currently large interest in the literature to use higher spatial resolution air quality model output to reprocess satellite retrievals. Therefore, these results will be important for others working on this.

*Expected FNR information from TEMPO:*

*- What is the expected signal ratio of Tempo compared to TROPOMI for NO2 and HCHO? Can we expect an improvement of the HCHO noise?*

In order to shorten the paper and provide more focus on the major results/conclusions, we have decided to remove this section of the paper. The authors feel that the TEMPO section does not fit well with the rest of the study and would be more appropriate in a different publication.

*- It would be interesting to show the diurnal variation of NO2 and HCHO from the TEMPO simulations.*

See our comment above about the removal of the synthetic TEMPO data section.

*- Line 777 and figure 7b and 7c. Not clear if retrieved OMI and TROPOMI are shown (line 777) or only synthetic TEMPO data averaged at the different spatial resolutions. It should be possible to show real data for OMI and TROPOMI in 2020.*

See our comment above about the removal of the synthetic TEMPO data section.

*Conclusion*

*Line 831: Please comment on the spatial and temporal resolution allowed by the OMI datasets. This is important for trend studies.*

We agree with the reviewer that this is an interesting problem for trend studies. However, defining exact temporal and spatial resolutions allowed by OMI or TROPOMI for these types of analysis is not the focus of our current study. In order to address this comment, we calculated daily mean tropospheric column quantities of $NO_2$, HCHO, and FNRs from both satellites and airborne data for the entire LISTOS domain and within 0.35 degrees of the NYC city center (identified as the emission source region) to calculate daily correlation statistics. The following text was added to Sect. 3.4.2 of the updated manuscript to summarize this evaluation and results "Given the limited spatiotemporal data coverage provided by the LISTOS campaign, a robust understanding of the temporal capabilities of OMI and TROPOMI to retrieve FNRs is not possible. LEO satellites obtain, at best, a single snapshot of both HCHO and $NO_2$ each day, so one could only hope to obtain daily variability of FNRs from these spaceborne systems. To determine whether OMI and TROPOMI could capture the variability of the daily mean tropospheric column quantities of $NO_2$, HCHO, and FNRs over the entire LISTOS domain from airborne data, we compared these daily mean values from NASA OMI, QA4ECV OMI, and TROPOMI to the airborne observations. For NASA OMI, daily correlation ($R^2$) values were 0.85 (p = 0.001), 0.58 (p = 0.03), and 0.26 (p = 0.20) for $NO_2$, HCHO, and FNRs, respectively. For QA4ECV OMI, daily correlation values were 0.85 (p = 0.001), 0.80 (p = 0.002), and 0.47 (p = 0.06) for $NO_2$, HCHO, and FNRs, respectively. For TROPOMI, daily correlation values were 0.92 (p = <0.001), 0.85 (p = <0.001), and 0.41 (p = 0.03) for $NO_2$, HCHO, and FNRs, respectively. All daily correlation statistics for HCHO and $NO_2$ were significant to a 95% confidence interval and suggest that both OMI and TROPOMI can capture the overall inter-daily magnitudes of FNR indicator species. However, only TROPOMI could observe the daily variability of domain-wide FNRs within a 95% confidence interval. This suggests that unresolved errors in either HCHO or $NO_2$ retrievals (the analysis from this study suggests uncertainty in HCHO are driving FNR bias variability) from OMI, using both the NASA and QA4ECV algorithms, are too large to confidently capture the inter-daily variability in FNRs.

The same analysis was conducted for NASA and QA4ECV OMI except just for retrievals near the large anthropogenic source regions in NYC (within 0.35 degrees of the city center) where relative errors due to satellite retrievals for FNR calculations were the lowest (see Fig. 6). Daily correlation ($R^2$) values for FNR retrievals near the source region of NYC for NASA OMI (0.13; p-value = 0.39) were reduced compared to domain-wide means and QA4ECV OMI (0.66; p-value = 0.01) correlations were improved near the source region of NYC. Indicator species correlation values from NASA OMI were degraded compared to the domain-wide analysis suggesting that this satellite product may not be able to capture inter-daily variability of FNRs even in large source regions. However, this analysis suggests that QA4ECV OMI data has the capability to retrieve daily variability of FNRs in large emission regions such as NYC to a statistically significant level. Overall, TROPOMI retrievals at both fine and coarse spatial resolutions evaluated in this study are able to capture daily variability of tropospheric FNRs over the entire domain and emission source regions better compared to OMI products.".

To further address this comment, we added an entire section (Sect. 3.4.2) in the updated manuscript which describes the capabilities of OMI and TROPOMI to observe spatial and temporal variability of FNRs during LISTOS. Furthermore, the following text was added to this section discussing daily- versus monthly-averaging OMI FNR data: "Recent studies have shown that averaging OMI data (especially HCHO retrievals) for longer temporal periods can reduce the noise and uncertainty in this data product. For example, in the recent paper by Souri et al. (2022a), it was shown that unresolved errors in OMI HCHO can be reduced in monthly-averages compared to daily retrievals by ~33% while there was little improvement in uncertainty statistics of $NO_2$ retrievals from OMI. However, recent studies (e.g., Schroeder et al., 2017) have also shown that for trend studies, monthly-averaging column FNR data can mask FNR temporal gradients that exist within that period. This could hinder the results of trend studies of pollution conditions on $O_3$ exceedance days, and days of lower pollution, which is a primary purpose of using satellite column FNR data.".

*Line 860. The statements made on the new version of the NASA OMI HCHO product appear to be optimistic. The SNR of the retrievals is primarily determined by the SNR of the instrument. Please be more cautious, especially since no publication can support the statements.*

We agree with the reviewer that this comment could be viewed as too optimistic without data analysis to support it. We have removed it from the conclusion section.

*Line 866-867: This does not seem so clear in the paper that "using the WRF-CMAQ-predicted a priori information, resulted in highly accurate retrievals of FNRs". All L2 products used in the study also results in median biases lower than 0.5 for FNRs.*

The reviewer is correct, and this sentence has been updated to reflect that the systematic bias is similar to the operational satellite products. However, other aspects of the statistical analysis of the reprocessed OMI retrievals were improved such as the correlations and RMSE values. This is discussed in more detail in the conclusion section of the updated manuscript.

*Line 871-872: This sentence is misleading. The need for accurate shape factors is not only for OMI retrievals. It should be even more important for TROPOMI and TEMPO because of their finer spatial resolution.*

This sentence has been updated to include TROPOMI.

**References**

[revised manuscript text omitted]

---

## Author Response (AR2)

We thank the editor and additional reviewer for the detailed evaluation of the paper. We have gone through and addressed their comments as described below. All editor/reviewer comments are presented in italic font while the author responses are displayed in standard font.

*Your revised manuscript addresses the main points raised by the reviewers, and is therefore acceptable for publication in AMT. However, the manuscript is very lengthy, and I find it tough to go through, even while being familiar with the topic. Therefore I strongly recommend that you shorten the manuscript wherever possible. There is quite a bit of repetition in the text, for example when you extensively quote the outcome (including uncertainties) for all the comparisons, when these can also be seen from the tables or figures. The text reaches a total of almost 1000 lines, and I think this can and should be reduced by at least 20%, so that readers can more quickly digest the essence of your work. The work basically does not justify 1000 lines. One of the reviewers, while positive about your revision recommended a useful opportunity to make the paper more concise: "I'd suggest the authors combine the discussions of biases and uncertainty, so that readers could have a wholistic picture about the performance of each satellite retrieval".*

The authors have done our best to reduce the length of the text. From the title to the end of the Data Availability section was 988 lines and is now 801 lines. This is a 19% reduction in length. We have tried to reduce the repetition in the text as much as possible and have combined the systematic bias and uncertainty discussion into single sections.

*One other reviewer had two useful comments, which should also be addressed:*

*(1) The abstract focuses very much on absolute bias numbers while the analysis (e.g. the regression slopes far from unity) indicate that these have limited meaning (many of the effects being multiplicative). Perhaps consider reporting %-biases or slopes (instead or additionally)?*

We now discuss the systematic bias and uncertainty value primarily using normalized mean bias percentage and linear regression slopes in the abstract.

*(2) Section 2.4: Agreed on the importance of the assumed vertical profile in the satellite retrievals, and the value of homogenizing these, but what about the profile used in the retrievals of the GeoTASO and GCAS tropospheric columns? In 3.3, you write "GeoTASO and GCAS retrievals were not reprocessed in order to have a consistent reference data set for satellite evaluation." I do not understand that argument. Why would reprocessing them mean they are no longer consistent (with what?)? There would of course be a concern about independence, but I believe it would have been a meaningful exercise nonetheless, unless I miss a strong argument against it. Actually, unless I missed it, little detail is given about the profile that underpins the airborne retrievals. Some text on this would be appreciated.*

The text in this section has been updated to "GeoTASO and GCAS retrievals were not reprocessed in order to have a consistent reference data set for the evaluation of the standard and reprocessed satellite retrievals. While reprocessing the airborne data with the higher spatial resolution model output would in itself be interesting (as done in Judd et al., 2020), the direct evaluation of the improvements in the reprocessed satellite data compared to the standard

retrieval would not be possible.". We wanted the GeoTASO and GCAS data to remain constant during the evaluation of both the standard and reprocessed satellite products to determine the potential improvements in the reprocessed satellite data. Thus, we refer to this as keeping the GeoTASO/GCAS consistent.

*Specific comments from the editor*:

*Throughout the manuscript you use two words for many concepts, where only one should be used. Example: on line 30 you state "clean/background". Just use "clean". There many such cases.*

We now use the term "clean" throughout the manuscript.

*L34: this sentence should be phrased such it makes clear that the number is a campaign average NO2 uncertainty. Now it could be interpreted as a generic finding, which is not the case. Same for the HCHO uncertainty on L42-43.*

We have identified these uncertainty values as campaign-averaged in the abstract.

*The abstract is too long - I suggest to shorten it by 30%.*

The abstract has been reduced by ~35%.

*L141: the equator overpass time of OMI is 13:30 hrs.*

The equatorial overpass time for Aura-OMI is 13:45 (https://earth.esa.int/eogateway/missions/aura/description, https://aura.gsfc.nasa.gov/scinst.html, https://earth.esa.int/eogateway/missions/aura).

*L167: please double check that the NASA NO2 algorithm uses the OMI O2-O2 cloud parameters. They have been using the Raman cloud product from OMI (Joiner-papers) in the past, and I doubt that this has changed. Regardless I think the O2-O2 product citation should be Acarreta (2002) or Veefkind (2016), not Vassilkov.*

Originally, the O2-O2 algorithm was developed by KNMI, and the representative papers are in fact Acarreta et al. (2004) and Veefkind et al. (2016) (KNMI product is named OMCLDO2). Vasilkov et al. (2018) produced an OMI cloud product using the O2-O2 algorithm and named it OMCLDO2N (or OMCLDO2N). The previous version of the NASA NO2 product (V3.1) used OMCLDO2, but then they replaced it with OMCLDO2N (NASA version 4.0 used in our study). In short, NASA produced OMCLDO2N for use in OMI (Vasilkov et al., 2018), which is broadly similar to the OMCLDO2 but has a number of differences.

We have added the reference to Acarreta et al. (2004) and Veefkind et al. (2016) when we first discuss the O2-O2 cloud parameter.

*L224: it is useful to remind the scientific community that v2.3 of TROPOMI NO2 has indeed improved compared to previous versions. The main reason for this is -better cloud pressures via FRESCO+ wide- explained and demonstrated in Riess et al. (2022), and it would be appropriate to cite that study here (along with van Geffen et al. (2022)) here.*

See our response to the comment about L442-444. The study by Riess et al. (2022) is now referenced.

*L341: qa_values should not be zero or < threshold, but > threshold. Please correct*

The wording for this discussion has been updated accordingly.

*L360-367: these lines can be removed*

This opening paragraph has been removed.

*L442-444: here the main reason why v2.3 has improved could be brought up as explained in Riess et al. (2022)*

The text has been updated to read: "It should be noted that the TROPOMI low bias in tropospheric column $NO_2$ is improved with the newer retrieval algorithm used in this study compared to early versions of the data product (e.g., v1.2.2 had a campaign-averaged median low bias of $-1.3\pm4.0 \times 10^{15}$ molecules $cm^{-2}$) primarily due to better cloud pressure input data (FRESCO+ wide) now used in TROPOMI retrievals (Riess et al., 2022)".

*L445: that NO2 (and all spaceborne tropospheric DOAS retrievals) have a magnitude dependence is well-known since the early works of Martin et al (2002) and Boersma et al. (2004). Those studies showed that the uncertainties in the AMF drive this magnitude dependency.*

This has now been acknowledged with the following text: "This magnitude dependence has been shown to be driven by uncertainties AMF values used in the retrievals (Martin et al., 2002; Boersma et al., 2004).".

*Caption Figure 6: is this Figure showing the FNR derived from the NASA scaled or NASA standard products? This should be clarified in the caption.*

This figure caption now states "Campaign-averaged relative error in FNR products from standard NASA OMI, QA4ECV OMI, and TROPOMI retrievals …"

*References*

*Riess, T. C. V. W., Boersma, K. F., van Vliet, J., Peters, W., Sneep, M., Eskes, H., and van Geffen, J.: Improved monitoring of shipping NO2 with TROPOMI: decreasing NOx emissions in European seas during the COVID-19 pandemic, Atmos. Meas. Tech., 15, 1415–1438, https://doi.org/10.5194/amt-15-1415-2022, 2022.*

*Please go through the manuscript one more time, address the above comments, and make the paper more concise, and then resubmit. Thanks for an informative paper.*

---

## Author Response (AR3)

We thank the editor for their final approval of the manuscript. We have addressed the final comments as described below. All editor comments are presented in italic font while the author responses are displayed in standard font.

*With the next revision, please add a section "Competing interests" (before the Acknowledgements) with the following text: John Sullivan is a member of the editorial board of Atmospheric Measurement Techniques. You uploaded a new supplement file. If you applied any changes to the previous supplement version, a tracked changes file is mandatory. If applicable, I kindly ask you to include the tracked-changes of your manuscript together with the tracked-changes of your supplement in one PDF and upload them as the "Author's tracked changes". Your reference list includes works "in review". Such works can be cited upon submission if being available to the reviewers. They should not be cited in the final, accepted manuscript, unless published, accepted for publication, or available as preprint with a DOI.*

- We have added the following Competing Interests statement: "John Sullivan is a member of the editorial board of Atmospheric Measurement Techniques.".
- The previous minor changes to the Supplemental Information document have been included in the tracked changes for the final manuscript version.
- We have added the final version of the reference for all previously "in review" citations.